# When Do Transformers Shine in RL?
# Decoupling Memory from Credit Assignment

**Tianwei Ni**
Mila, Université de Montréal
`tianwei.ni@mila.quebec`

**Michel Ma**
Mila, Université de Montréal
`michel.ma@mila.quebec`

**Benjamin Eysenbach**
Princeton University
`eysenbach@princeton.edu`

**Pierre-Luc Bacon**
Mila, Université de Montréal
`pierre-luc.bacon@mila.quebec`

## Abstract

Reinforcement learning (RL) algorithms face two distinct challenges: learning effective representations of past and present observations, and determining how actions influence future returns. Both challenges involve modeling long-term dependencies. The Transformer architecture has been very successful to solve problems that involve long-term dependencies, including in the RL domain. However, the underlying reason for the strong performance of Transformer-based RL methods remains unclear: is it because they learn effective memory, or because they perform effective credit assignment? After introducing formal definitions of memory length and credit assignment length, we design simple configurable tasks to measure these distinct quantities. Our empirical results reveal that Transformers can enhance the memory capability of RL algorithms, scaling up to tasks that require memorizing observations 1500 steps ago. However, Transformers do not improve long-term credit assignment. In summary, our results provide an explanation for the success of Transformers in RL, while also highlighting an important area for future research and benchmark design. Our code is open-sourced[1].

## 1 Introduction

In recent years, Transformers (Vaswani et al., 2017; Radford et al., 2019) have achieved remarkable success in domains ranging from language modeling to computer vision. Within the RL community, there has been excitement around the idea that large models with attention architectures, such as Transformers, might enable rapid progress on challenging RL tasks. Indeed, prior works have shown that Transformer-based methods can achieve excellent results in offline RL (Chen et al., 2021; Janner et al., 2021; Lee et al., 2022), online RL (Parisotto et al., 2020; Lampinen et al., 2021; Zheng et al., 2022; Melo, 2022; Micheli et al., 2022; Robine et al., 2023), and real-world tasks (Ouyang et al., 2022; Ichter et al., 2022) (see Li et al. (2023); Agarwal et al. (2023) for the recent surveys).

However, the underlying reasons why Transformers achieve excellent results in the RL setting remain a mystery. Is it because they learn better representations of sequences, akin to their success in computer vision and NLP tasks? Alternatively, might they be internally implementing a learned algorithm, one that performs better credit assignment than previously known RL algorithms? At the core of these questions is the fact that RL, especially in partially observable tasks, requires two distinct forms of temporal reasoning: (working) **memory** and (temporal) **credit assignment**. Memory refers to the ability to recall a distant past event at the current time (Blankenship, 1938; Dempster, 1981),

---

[1] `https://github.com/twni2016/Memory-RL`

while credit assignment is the ability to determine *when* the actions that deserve current credit occurred (Sutton, 1984). These two concepts are also loosely intertwined – learning what bits of history to remember depends on future rewards, and learning to assign current rewards to previous actions necessitates some form of (episodic) memory (Zilli and Hasselmo, 2008; Gershman and Daw, 2017).

Inspired by the distinct nature of memory and credit assignment in temporal dependencies, we aim to understand which of these two concepts Transformers address in the context of RL. Despite empirical success in prior works, directly answering the question is challenging due to two issues. First, many of the benchmarks used in prior works, such as Atari (Bellemare et al., 2013) and MuJoCo (Todorov et al., 2012), require minimal memory and short-term credit assignment, because they closely resemble MDPs and their rewards are mostly immediate. Other benchmarks, such as Key-to-Door (Hung et al., 2018; Mesnard et al., 2021), entangle medium-term memory and credit assignment. Second, there is a lack of rigorous definition of memory and credit assignment in RL (Osband et al., 2020). The absence of quantifiable measures creates ambiguity in the concept of "long-term" often found in prior works.

In this paper, we address both issues concerning memory and credit assignment to better understand Transformers in RL. First, we provide a mathematical definition of memory lengths and credit assignment lengths in RL, grounded in common understanding. We distinguish the memory lengths in reward, transition, policy, and value, which relate to the minimal lengths of recent histories needed to preserve the corresponding quantities. Crucially, our main theoretical result reveals that the memory length of an optimal policy can be upper bounded by the reward and transition memory lengths. For credit assignment, we adopt the forward view and define its length as the minimal number of future steps required for a greedy action to start outperforming any non-greedy counterpart.

The definitions provided are not only quantitative in theory but also actionable for practitioners, allowing them to analyze the memory and credit assignment requirements of many existing tasks. Equipped with these tools, we find that many tasks designed for evaluating memory also evaluate credit assignment, and vice versa, as demonstrated in Table 1. As a solution, we introduce toy examples called Passive and Active T-Mazes that decouple memory and credit assignment. These toy examples are configurable, enabling us to perform a scalable unit-test of a sole capability, in line with best practice (Osband et al., 2020).

Lastly, we evaluate memory-based RL algorithms with LSTMs or Transformers on our configurable toy examples and other challenging tasks to address our research question. We find that Transformers can indeed boost long-term memory in RL, scaling up to memory lengths of 1500. However, Transformers do not improve long-term credit assignment, and struggle with data complexity in even short-term dependency tasks. While these results suggest that practitioners might benefit from using Transformer-based architectures in RL, they also highlight the importance of continued research on core RL algorithms. Transformers have yet to replace RL algorithm designers.

## 2 Measuring Temporal Dependencies in RL

**MDPs and POMDPs.** In a partially observable Markov decision process (POMDP) $\mathcal{M}_O = (\mathcal{O}, \mathcal{A}, P, R, \gamma, T)$[2], an agent receives an observation $o_t \in \mathcal{O}$ at step $t \in \{1, \ldots, T\}$, takes an action $a_t \in \mathcal{A}$ based on the observed history $h_{1:t} := (o_{1:t}, a_{1:t-1}) \in \mathcal{H}_t$[3], and receives a reward $r_t \sim R_t(h_{1:t}, a_t)$ and the next observation $o_{t+1} \sim P(\cdot \mid h_{1:t}, a_t)$. The initial observation $h_{1:1} := o_1$ follows $P(o_1)$. The total horizon is $T \in \mathbb{N}^+ \cup \{+\infty\}$ and the discount factor is $\gamma \in [0, 1]$ (less than 1 for infinite horizon). In a Markov decision process (MDP) $\mathcal{M}_S = (\mathcal{S}, \mathcal{A}, P, R, \gamma, T)$, the observation $o_t$ and history $h_{1:t}$ are replaced by the state $s_t \in \mathcal{S}$. In the most generic case, the agent is composed of a policy $\pi(a_t \mid h_{1:t})$ and a value function $Q^\pi(h_{1:t}, a_t)$. The optimal value function $Q^*(h_{1:t}, a_t)$ satisfies $Q^*(h_{1:t}, a_t) = \mathbb{E}[r_t \mid h_{1:t}, a_t] + \gamma \mathbb{E}_{o_{t+1} \sim P(\mid h_{1:t}, a_t)}[\max_{a_{t+1}} Q^*(h_{1:t+1}, a_{t+1})]$ and induces a deterministic optimal policy $\pi^*(h_{1:t}) = \operatorname{argmax}_{a_t} Q^*(h_{1:t}, a_t)$. Let $\Pi_\mathcal{M}^*$ be the space of all optimal policies for a POMDP $\mathcal{M}$.

---

[2]The classic notion of POMDPs (Kaelbling et al., 1998) includes a state space, but we omit it because we assume that it is unknown.

[3]For any $k \in \{1, \ldots, t\}$, let $o_{t-k:t}$ denote the sequence $(o_{t-k}, \ldots, o_t)$, and similarly for $a_{t-k:t}$. In particular, let $h_{t:t} = o_t$ to align with the MDP setting, and let $h_{t':t} = \emptyset$ for $t' > t$ and $h_{t':t} = h_{1:t}$ for $t' < 1$.

Below, we provide definitions for context-based policy and value function, as well as the sum of $n$-step rewards given a policy. These concepts will be used in defining memory and credit assignment lengths.

**Context-based policy and value function.** A context-based policy $\pi$ takes the recent $l_{\mathrm{ctx}}(\pi)$ observations and actions as inputs, expressed as $\pi(a_t \mid h_{t-l_{\mathrm{ctx}}(\pi)+1:t})$. Here, $l_{\mathrm{ctx}}(\pi) \in \mathbb{N}$ represents the **policy context length**. For simplicity, we refer to context-based policies as "policies". Let $\Pi_k$ be the space of all policies with a context length of $k$. Markovian policies form $\Pi_1$, while $\Pi_T$, the largest policy space, contains an optimal policy. Recurrent policies, recursively taking one input at a time, belong to $\Pi_\infty$; while Transformer-based policies have a limited context length. **Context-based value function** $Q_n^\pi$ of a POMDP $\mathcal{M}$ is the expected return conditioned on the past $n$ observations and actions, denoted as $Q_n^\pi(h_{t-n+1:t}, a_t) = \mathbb{E}_{\pi,\mathcal{M}}\left[\sum_{i=t} \gamma^{i-t} r_i \mid h_{t-n+1:t}, a_t\right]$. By definition, $Q^\pi(h_{1:t}, a_t) = Q_T^\pi(h_{1:t}, a_t), \forall h_{1:t}, a_t$.

**The sum of $n$-step rewards.** Given a POMDP $\mathcal{M}$ and a policy $\pi$, the expectation of the discounted sum of $n$-step rewards is denoted as $G_n^\pi(h_{1:t}, a_t) = \mathbb{E}_{\pi,\mathcal{M}}\left[\sum_{i=t}^{t+n-1} \gamma^{i-t} r_i \mid h_{1:t}, a_t\right]$. By definition, $G_1^\pi(h_{1:t}, a_t) = \mathbb{E}[r_t \mid h_{1:t}, a_t]$ (immediate reward), and $G_{T-t+1}^\pi(h_{1:t}, a_t) = Q^\pi(h_{1:t}, a_t)$ (value).

## 2.1 Memory Lengths

In this subsection, we introduce the memory lengths in the common components of RL, including the reward, transition, policy, and value functions. Then, we will show the theoretical result of the relation between these lengths.

The reward (or transition) memory length of a POMDP quantifies the temporal dependency distance between *current expected reward* (or *next observation distribution*) and the most distant observation.

**Definition 1.A (Reward memory length $m_{\mathrm{reward}}^{\mathcal{M}}$).** *For a POMDP $\mathcal{M}$, $m_{\mathrm{reward}}^{\mathcal{M}}$ is the smallest $n \in \mathbb{N}$ such that the expected reward conditioned on recent $n$ observations is the same as the one conditioned on full history,* i.e., $\mathbb{E}[r_t \mid h_{1:t}, a_t] = \mathbb{E}[r_t \mid h_{t-n+1:t}, a_t], \forall t, h_{1:t}, a_t$.

**Definition 1.B (Transition memory length $m_{\mathrm{transit}}^{\mathcal{M}}$).** *For a POMDP $\mathcal{M}$, $m_{\mathrm{transit}}^{\mathcal{M}}$ is the smallest $n \in \mathbb{N}$ such that next observations are conditionally independent of the rest history given the past $n$ observations and actions,* i.e., $o_{t+1} \perp\!\!\!\perp h_{1:t}, a_t \mid h_{t-n+1:t}, a_t$.

Next, policy memory length is defined based on the intuition that while a policy may take a long history of observations as its context, its action distribution only depends on recent observations. Therefore, policy memory length represents the temporal dependency distance between current *action* and the most distant observation.

**Definition 1.C (Policy memory length $l_{\mathrm{mem}}(\pi)$).** *The policy memory length $l_{\mathrm{mem}}(\pi)$ of a policy $\pi$ is the minimum horizon $n \in \{0, \ldots, l_{\mathrm{ctx}}(\pi)\}$ such that its actions are conditionally independent of the rest history given the past $n$ observations and actions,* i.e., $a_t \perp\!\!\!\perp h_{t-l_{\mathrm{ctx}}(\pi)+1:t} \mid h_{t-n+1:t}$.

Lastly, we define the value memory length of a policy, which measures the temporal dependency distance between the current *value* and the most distant observation.

**Definition 1.D (Value memory length of a policy $l_{\mathrm{value}}^{\mathcal{M}}(\pi)$).** *A policy $\pi$ has its value memory length $l_{\mathrm{value}}^{\mathcal{M}}(\pi)$ as the smallest $n$ such that the context-based value function is equal to the value function,* i.e., $Q_n^\pi(h_{t-n+1:t}, a_t) = Q^\pi(h_{1:t}, a_t), \forall t, h_{1:t}, a_t$.

Now we prove that the policy and value memory lengths of optimal policies can be upper bounded by the reward and transition memory lengths (Proof in Appendix A).

**Theorem 1 (Upper bounds of memory lengths for optimal policies).** *For optimal policies with the shortest policy memory length, their policy memory lengths are upper bounded by their value memory lengths, which are further bounded by the maximum of reward and transition memory lengths.*

$$l_{\mathrm{mem}}(\pi^*) \leq l_{\mathrm{value}}^{\mathcal{M}}(\pi^*) \leq \max(m_{\mathrm{reward}}^{\mathcal{M}}, m_{\mathrm{transit}}^{\mathcal{M}}) := m^{\mathcal{M}}, \quad \forall \pi^* \in \operatorname*{argmin}_{\pi \in \Pi_{\mathcal{M}}^*}\{l_{\mathrm{mem}}(\pi)\} \quad (1)$$

Thm. 1 suggests that the constant of $m^{\mathcal{M}}$ can serve as a proxy to analyze the policy and value memory lengths required in a task, which are often challenging to directly compute, as we will demonstrate in Sec. 3.[4]

## 2.2 Credit Assignment Lengths

Classically, temporal credit assignment is defined with a backward view (Minsky, 1961; Sutton, 1984) – how previous actions contributed to current reward. The backward view is specifically studied through gradient back-propagation in supervised learning (Lee et al., 2015; Ke et al., 2018). In this paper, we consider the forward view in reinforcement learning – how current action will influence future rewards. Specifically, we hypothesize that the credit assignment length $n$ of a policy is how long the greedy action at the current time-step *starts to* make a difference to the sum of future $n$-step rewards.

Before introducing the formal definition, let us recall the common wisdom on dense-reward versus sparse-reward tasks. Dense reward is generally easier to solve than sparse reward if all other aspects of the tasks are the same. One major reason is that an agent can get immediate reward feedback in dense-reward tasks, while the feedback is delayed or even absent in sparse-reward tasks. In this view, solving sparse reward requires long-term credit assignment. Our definition will reflect this intuition.

**Definition 2.A** (**Credit assignment length of a history given a policy** $c(h_{1:t}; \pi)$)**.** *Given a policy $\pi$ in a task, the credit assignment length of a history $h_{1:t}$ quantifies the minimal temporal distance $n$[5], such that a greedy action $a_t^*$ is strictly better than any non-greedy action $a_t'$ in terms of their $n$-step rewards $G_n^\pi$. Formally, let $A_t^* := \mathrm{argmax}_{a_t} Q^\pi(h_{1:t}, a_t)$[6], define*

$$c(h_{1:t}; \pi) := \min_{1 \leq n \leq T-t+1} \{n \mid \exists a_t^* \in A_t^*, \text{s.t. } G_n^\pi(h_{1:t}, a_t^*) > G_n^\pi(h_{1:t}, a_t'), \forall a_t' \notin A_t^*\} \quad (2)$$

**Definition 2.B** (**Credit assignment length of a policy** $c(\pi)$ **and of a task** $c^{\mathcal{M}}$)**.** *Credit assignment length of a policy $\pi$ represents the worst case of credit assignment lengths of all visited histories. Formally, let $d_\pi$ be the history occupancy distribution of $\pi$, define $c(\pi) := \max_{h_{1:t} \in \mathrm{supp}(d_\pi)} c(h_{1:t}; \pi)$.*

*Further, we define the credit assignment length of a task $\mathcal{M}$, denoted as $c^{\mathcal{M}}$, as the minimal credit assignment lengths over all optimal policies[7], i.e., $c^{\mathcal{M}} := \min_{\pi^* \in \Pi_{\mathcal{M}}^*} c(\pi^*)$.*

As an illustrative example, consider the extreme case of a sparse-reward MDP where the reward is only provided at the terminal time step $T$. In this environment, the credit assignment length of any policy $\pi$ is $c(\pi) = T$. This is because the credit assignment length of the initial state $s_1$ is always $c(s_1; \pi) = T$, as the performance gain of any greedy action at $s_1$ is only noticeable after $T$ steps.

Our definition of credit assignment lengths has ties with the temporal difference (TD) learning algorithm (Sutton, 1988). With the tabular TD(0) algorithm, the value update is given by $V(s_t) \leftarrow V(s_t) + \alpha(r_t + \gamma V(s_{t+1}) - V(s_t))$. This formula back-propagates the information of current reward $r_t$ and next value $V(s_{t+1})$ to current value $V(s_t)$. When we apply the TD(0) update backwards in time for $n$ steps along a trajectory, we essentially back-propagate the information of $n$-step rewards to the current value. Thus, the credit assignment length of a state measures the number of TD(0) updates we need to take a greedy action for that state.

## 3 Environment Analysis of Memory and Credit Assignment Lengths

By employing the upper bound $m^{\mathcal{M}}$ on memory lengths (Thm. 1) and credit assignment length $c^{\mathcal{M}}$ (Def. 2.B), we can perform a quantitative analysis of various environments. These environments include both abstract problems and concrete benchmarks. In this section, we aim to address two questions: (1) Do prior environments actually require long-term memory or credit assignment? (2) Can we disentangle memory from credit assignment?

---

[4]Thm. 1 also applies to MDPs, where it provides a quick proof of the well-known fact that optimal policies and values of MDPs can be Markovian: $m_{\mathrm{reward}}^{\mathcal{M}}, m_{\mathrm{transit}}^{\mathcal{M}}$ are both at most 1, so the policy and value memory lengths are also at most 1.

[5]The $n$ is guaranteed to exist because a greedy action must be the best in $T - t + 1$-step rewards (*i.e.* $Q$-value).

[6]If $A_t^* = \mathcal{A}$, *i.e.* all actions are greedy, then the set of $n$ contains all integers between 1 and $T - t + 1$.

[7]If there are multiple optimal policies, $c(\pi^*)$ may be not unique. See Appendix A for an example.

Table 1: ***Estimated*** **memory and credit assignment lengths required in prior tasks**, as defined in Sec. 2. The top block shows tasks designed for memory, while the bottom one shows tasks designed for credit assignment. All tasks have a horizon of $T$ and a discount factor of $\gamma \in (0, 1)$. The $T$ column shows the largest value used in prior work. The terms "long" and "short" are relative to horizon $T$. We *manually* estimate these lengths from task definition and mark tasks *purely* evaluating long-term memory or credit assignment in black.

| | Task $\mathcal{M}$ | $T$ | $l_{\mathrm{mem}}(\pi^*)$ | $m^{\mathcal{M}}$ | $c^{\mathcal{M}}$ |
|---|---|---|---|---|---|
| **Memory** | Reacher-pomdp (Yang and Nguyen, 2021) | 50 | long | long | short |
| | Memory Cards (Esslinger et al., 2022) | 50 | long | long | 1 |
| | TMaze Long (Noise) (Beck et al., 2019) | 100 | $T$ | $T$ | 1 |
| | Memory Length (Osband et al., 2020) | 100 | $T$ | $T$ | 1 |
| | Mortar Mayhem (Pleines et al., 2023) | 135 | long | long | $\leq 25$ |
| | Autoencode (Morad et al., 2023) | 311 | $T$ | $T$ | 1 |
| | Numpad (Parisotto et al., 2020) | 500 | long | long | short |
| | PsychLab (Fortunato et al., 2019) | 600 | $T$ | $T$ | short |
| | Passive Visual Match (Hung et al., 2018) | 600 | $T$ | $T$ | $\leq 18$ |
| | Repeat First (Morad et al., 2023) | 831 | 2 | $T$ | 1 |
| | Ballet (Lampinen et al., 2021) | 1024 | $\geq 464$ | $T$ | short |
| | Passive Visual Match (Sec. 5.1; Our experiment) | 1030 | $T$ | $T$ | $\leq 18$ |
| | $17^2$ MiniGrid-Memory (Chevalier-Boisvert et al., 2018) | 1445 | $\leq 51$ | $T$ | $\leq 51$ |
| | Passive T-Maze (Eg. 1; Ours) | 1500 | $T$ | $T$ | 1 |
| | $15^2$ Memory Maze (Pasukonis et al., 2022) | **4000** | long | long | $\leq 225$ |
| | HeavenHell (Esslinger et al., 2022) | 20 | $T$ | $T$ | $T$ |
| | T-Maze (Bakker, 2001) | 70 | $T$ | $T$ | $T$ |
| | Goal Navigation (Fortunato et al., 2019) | 120 | $T$ | $T$ | $T$ |
| | T-Maze (Lambrechts et al., 2021) | 200 | $T$ | $T$ | $T$ |
| | Spot the Difference (Fortunato et al., 2019) | 240 | $T$ | $T$ | $T$ |
| | PyBullet-P benchmark (Ni et al., 2022) | 1000 | 2 | 2 | short |
| | PyBullet-V benchmark (Ni et al., 2022) | 1000 | short | short | short |
| **Credit Assignment** | Umbrella Length (Osband et al., 2020) | 100 | 1 | 1 | $T$ |
| | Push-r-bump (Yang and Nguyen, 2021) | 50 | long | long | long |
| | Key-to-Door (Raposo et al., 2021) | 90 | short | $T$ | $T$ |
| | Delayed Catch (Raposo et al., 2021) | 280 | 1 | $T$ | $T$ |
| | Active T-Maze (Eg. 2; Ours) | 500 | $T$ | $T$ | $T$ |
| | Key-to-Door (Sec. 5.2; Our experiment) | 530 | short | $T$ | $T$ |
| | Active Visual Match (Hung et al., 2018) | 600 | $T$ | $T$ | $T$ |
| | Episodic MuJoCo (Ren et al., 2022) | **1000** | 1 | $T$ | $T$ |

## 3.1 How Long Do Prior Tasks Require Memory and Credit Assignment?

To study memory or credit assignment in RL, prior works propose abstract problems and evaluate their algorithms in benchmarks. A prerequisite for evaluating agent capabilities in memory or credit assignment is understanding the requirements for them in the tested tasks. Therefore, we collect representative problems and benchmarks with a similar analysis in Sec. 3.2, to figure out the quantities $m^{\mathcal{M}}$ related to memory lengths and $c^{\mathcal{M}}$ related to credit assignment length.

Regarding abstract problems, we summarize the analysis in Appendix B. For example, decomposable episodic reward assumes the reward is only given at the terminal step and can be decomposed into the sum of Markovian rewards. This applies to episodic MuJoCo (Ren et al., 2022) and Catch with delayed rewards (Raposo et al., 2021). We show that it has policy memory length $l_{\mathrm{mem}}(\pi^*)$ at most 1 while requiring $m^{\mathcal{M}}_{\mathrm{reward}}$ and $c^{\mathcal{M}}$ of $T$, indicating that it is really challenging in both memory and credit assignment. Delayed environments are known as difficult to assign credit (Sutton, 1984). We show that delayed reward (Arjona-Medina et al., 2019) or execution (Derman et al., 2021) or observation (Katsikopoulos and Engelbrecht, 2003) with $n$ steps all require $m^{\mathcal{M}}_{\mathrm{reward}}$ of $n$, while only delayed rewards and execution may be challenging in credit assignment.

As for concrete benchmarks, Table 1 summarizes our analysis, with detailed descriptions of these environments in Appendix B. First, some tasks designed for long-term memory (Beck et al., 2019; Osband et al., 2020; Hung et al., 2018; Yang and Nguyen, 2021; Esslinger et al., 2022) or credit

assignment (Osband et al., 2020) indeed *solely* evaluate the respective capability. Although some memory tasks such as Numpad (Parisotto et al., 2020) and Memory Maze (Pasukonis et al., 2022) do require long-term memory with short-term credit assignment, the exact memory and credit assignment lengths are hard to quantify due to the complexity. In contrast, our Passive T-Maze introduced in Sec. 3.2, provides a clear understanding of the memory and credit assignment lengths, allowing us to successfully scale the memory length up to 1500.

Conversely, other works (Bakker, 2001; Lambrechts et al., 2021; Esslinger et al., 2022; Hung et al., 2018; Raposo et al., 2021; Ren et al., 2022) actually entangle memory and credit assignment together, despite their intention to evaluate one of the two. Lastly, some POMDPs (Ni et al., 2022) only require short-term memory and credit assignment, which cannot be used for any capability evaluation.

## 3.2  Disentangling Memory from Credit Assignment

We can demonstrate that memory and credit assignment are independent concepts by providing two examples, all with a finite horizon of $T$ and no discount. The examples have distinct pairs of $(\max(r_{\mathcal{M}}, m_{\text{transit}}^{\mathcal{M}}), c^{\mathcal{M}})$: $(T,1), (T,T)$, respectively.

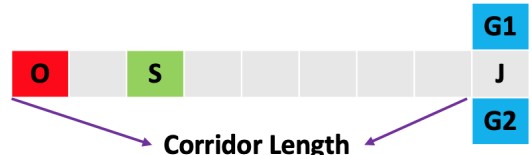

We adapt T-Maze (Fig. 1), a POMDP environment rooted in neuroscience (O'Keefe and Dostrovsky, 1971) and AI (Bakker, 2001; Osband et al., 2020), to allow different credit assignment lengths. Passive and active concepts are borrowed from Hung et al. (2018).

Figure 1: **T-Maze task.** In the passive version, the agent starts at the oracle state ("S" and "O" are the same state), while the active version requires that the agent navigate left from the initial state to collect information from the oracle state.

**Example 1** (**Passive T-Maze**). *T-Maze has a 4-direction action space $\{\mathcal{L}, \mathcal{R}, \mathcal{U}, \mathcal{D}\}$. It features a long corridor of length $L$, starting from (oracle) state $O$ and ending with (junction) state $J$, connecting two potential goal states $G_1$ and $G_2$. $O$ has a horizontal position of $0$, and $J$ has $L$. The agent observes null $N$ except at states $J, O, G_1, G_2$. The observation space $\{N, J, O, G_1, G_2\}$ is two-dimensional. At state $O$, the agent can observe the goal position $G \in \{G_1, G_2\}$, uniformly sampled at the beginning of an episode. The initial state is $S$. The transition is deterministic, and the agent remains in place if it hits the wall. Thus, the horizontal position of the agent $x_t \in \mathbb{N}$ is determined by its previous action sequence $a_{1:t-1}$.*

*In a Passive T-Maze $\mathcal{M}_{passive}$, $O = S$, allowing the agent to observe $G$ initially, and $L = T - 1$. Rewards are given by $R_t(h_{1:t}, a_t) = \frac{\mathbf{1}(x_{t+1} \geq t) - 1}{T - 1}$ for $t \leq T - 1$, and $R_T(h_{1:T}, a_T) = \mathbf{1}(o_{T+1} = G)$. The unique optimal policy $\pi^*$ moves right for $T - 1$ steps, then towards $G$, yielding an expected return of $1.0$. The optimal Markovian policy can only guess the goal position, ending with an expected return of $0.5$. The worst policy has an expected return of $-1.0$.*

**Example 2** (**Active T-Maze**). *In an Active T-Maze $\mathcal{M}_{active}$, $O$ is one step left of $S$ and $L = T - 2$. The unique optimal policy moves left to reach $O$, then right to $J$. The rewards $R_1 = 0$, $R_t(h_{1:t}, a_t) = \frac{\mathbf{1}(x_{t+1} \geq t-1) - 1}{T - 2}$ for $2 \leq t \leq T - 1$, and $R_T(h_{1:T}, a_T) = \mathbf{1}(o_{T+1} = G)$. The optimal Markovian and worst policies have the same expected returns as in the passive setting.*

$\mathcal{M}_{\text{passive}}$ is specially designed for testing memory only. To make the final decision, the optimal policy must recall $G$, observed in the first step. All the memory lengths are $T$ as the terminal reward depends on initial observation and observation at $J$ depends on the previous action sequence. The credit assignment length is 1 as immediate penalties occur for suboptimal actions. In contrast, $\mathcal{M}_{\text{active}}$ also requires a credit assignment length of $T$, as the initial optimal action $\mathcal{L}$ only affects $n$-step rewards when $n = T$. Passive and Active T-Mazes are suitable for evaluating the ability of pure memory, and both memory and credit assignment in RL, as they require little exploration due to the dense rewards before reaching the goal.

## 4  Related Work

Memory and credit assignment have been extensively studied in RL, with representative works shown in Table 1. The work most closely related to ours is `bsuite` (Osband et al., 2020), which

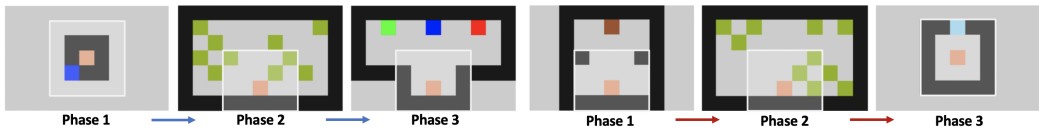

Figure 2: **Pixel-based tasks, Passive Visual Match (left) and Key-to-Door (right), evaluated in our experiments.** Each task has three phases, where we adjust the length of Phase 2 and keep the lengths of the others constant. The agent (in beige) only observes nearby grids (within white borders) and cannot pass through the walls (in black). In Passive Visual Match, the agent observes a randomized target color (blue in this case) in Phase 1, collects apples (in green) in Phase 2, and reaches the grid with the matching color in Phase 3. In Key-to-Door, the agent reaches the key (in brown) in Phase 1, collects apples in Phase 2, and reaches the door (in cyan) in Phase 3.

also disentangles memory and credit assignment in their benchmark. bsuite is also configurable with regard to the memory and credit assignment lengths to evaluate an agent's capability. Our work extends Osband et al. (2020) by providing formal definitions of memory and credit assignment lengths in RL for task design and evaluation. Our definitions enable us to analyze existing POMDP benchmarks.

Several prior works have assessed memory (Parisotto et al., 2020; Esslinger et al., 2022; Chen et al., 2022; Pleines et al., 2023; Morad et al., 2023) in the context of Transformers in RL. However, these prior methods use tasks that require (relatively short) context lengths between 50 and 831. In addition, there are two separate lines of work using Transformers for better credit assignment. One line of work (Liu et al., 2019; Ferret et al., 2020) has developed algorithms that train Transformers to predict rewards for return redistribution. The other line of work (Chen et al., 2021; Zheng et al., 2022) proposes return-conditioned agents trained on offline RL datasets. Both lines are beyond the scope of canonical online RL algorithms that purely rely on TD learning, which we focus on.

## 5   Evaluating Memory-Based RL Algorithms

In this section, we evaluate the memory and credit assignment capabilities of memory-based RL algorithms. Our first experiment aims to evaluate the memory abilities by disentangling memory from credit assignment, using our Passive T-Maze and the Passive Visual Match (Hung et al., 2018) tasks in Sec. 5.1. Next, we test the credit assignment abilities of memory-based RL in our Active T-Maze and the Key-to-Door (Raposo et al., 2021) tasks in Sec. 5.2. Passive Visual Match and Key-to-Door tasks (demonstrated in Fig. 2) can be viewed as pixel-based versions of Passive and Active T-Mazes, respectively. Lastly, we evaluate memory-based RL on standard POMDP benchmarks that only require short-term dependencies in Sec. 5.3, with a focus on their sample efficiency.

We focus on model-free RL agents that take observation and action *sequences* as input, which serves as a simple baseline that can achieve excellent results in many tasks (Ni et al., 2022). The agent architecture is based on the codebase from Ni et al. (2022), consisting of observation and action embedders, a sequence model, and actor-critic heads. We compare agents with LSTM and Transformer (GPT-2 (Radford et al., 2019)) architectures. For a fair comparison, we use the same hidden state size (128) and tune the number of layers (varying from $1, 2, 4$) for both LSTM and Transformer architectures. Since our tasks all require value memory lengths to be the full horizon, we set the *context lengths* to the full horizon $T$.

For the T-Maze tasks, we use DDQN (van Hasselt et al., 2016) with epsilon-greedy exploration as the RL algorithm to better control the exploration strategy. In pixel-based tasks, we use CNNs as observation embedders and SAC-Discrete (Christodoulou, 2019) as RL algorithm following Ni et al. (2022). We train all agents to convergence or at most 10M steps on these tasks. We provide training details and learning curves in Appendix C.

### 5.1   Transformers Shine in Pure Long-Term Memory Tasks

First, we provide evidence that Transformers can indeed enhance long-term memory in tasks that purely test memory, Passive T-Mazes. In Fig. 3 (left), Transformer-based agents consistently solve the task requiring memory lengths from 50 up to 1500. To the best of our knowledge, this achievement

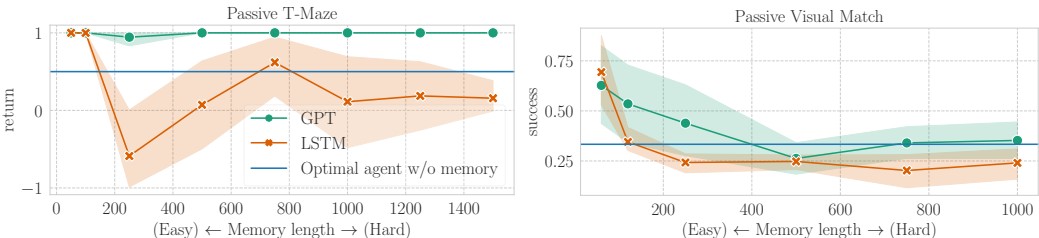

Figure 3: **Transformer-based RL outperforms LSTM-based RL in tasks (purely) requiring long-term memory. Left:** results in Passive T-Mazes with varying memory lengths from 50 to 1500; **Right:** results in Passive Visual Match with varying memory lengths from 60 to 1000. We also show the performance of the optimal Markovian policies. Each data point in the figure represents the final performance of an agent with the memory length indicated on the x-axis. All the figures in this paper show the mean and its 95% confidence interval over 10 seeds.

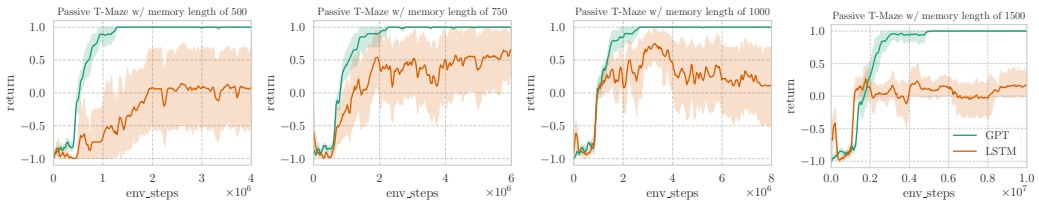

Figure 4: **Transformer-based agent can solve long-term memory low-dimensional tasks with good sample efficiency.** In Passive T-Maze with long memory length (from left to right: 500, 750, 1000, 1500), Transformer-based agent is significantly better than LSTM-based agent.

may well set a new record for the longest memory length of a task that an RL agent can solve. In contrast, LSTM-based agents start to falter at a memory length of 250, underperforming the optimal Markovian policies with a return of 0.5. Both architectures use a single layer, yielding the best performance. We also provide detailed learning curves for tasks requiring long memory lengths in Fig. 4. Transformer-based agent also shows much greater stability and sample efficiency across seeds when compared to their LSTM counterparts.

On the other hand, the pixel-based Passive Visual Match (Hung et al., 2018) has a more complex reward function: zero rewards for observing the color in Phase 1, immediate rewards for apple picking in Phase 2 requiring short-term dependency, and a final bonus for reaching the door of the matching color, requiring long-term memory. The success rate solely depends on the final bonus. The ratio of the final bonus w.r.t. the total rewards is inversely proportional to the episode length, as the number of apples increases over time in Phase 2. Thus, the low ratio makes learning long-term memory challenging for this task. As shown in Fig. 3 (right), Transformer-based agents slightly outperform LSTM-based ones[8] in success rates when the tasks require a memory length over 120, yet their success rates are close to that of the optimal Markovian policy (around 1/3). Interestingly, Transformer-based agents achieve much higher returns with more apples collected, shown in Fig. 5. This indicates Transformers can help short-term memory (credit assignment) in some tasks.

### 5.2 Transformers Enhance Temporal Credit Assignment in RL, but not Long-Term

Next, we present evidence that Transformers do not enhance long-term credit assignment for memory-based RL. As illustrated in Fig. 6 (left), in Active T-Mazes, by merely shifting the Oracle one step to the left from the starting position as compared to Passive T-Mazes, both Transformer-based and LSTM-based agents fail to solve the task when the credit assignment (and memory) length extends to just 250. The failure of Transformer-based agents on Active T-Mazes suggests the bottleneck of credit assignment. This is because they possess sufficient memory capabilities to solve this task (as evidenced by their success in Passive T-Mazes), and they can reach the junction state without exploration issues (as evidenced by their return of more than 0.0). A similar trend is observed in pixel-based Key-to-Door tasks (Raposo et al., 2021), where both Transformer-based and LSTM-

---

[8]It is worth noting that at the memory length of 1000, LSTM-based agents encountered NaN gradient issues in around 30% of the runs, which were not shown in the plots.

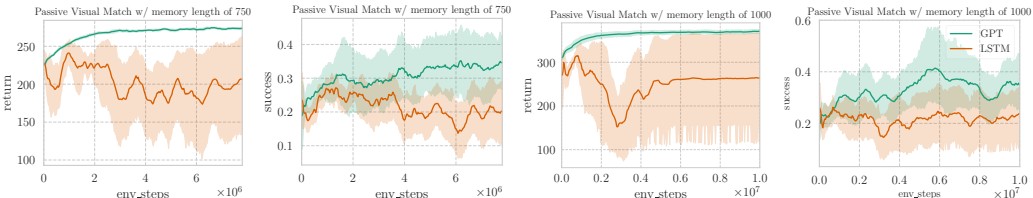

Figure 5: **Transformer-based agent is more sample-efficient in long-term memory high-dimensional tasks.** In Passive Visual Match with long memory length (750 on the left; 1000 on the right), Transformer-based agents show a slight advantage over LSTM-based agents in the memory capability (as measured by **success rate**), while being much higher in total rewards (as measured by **return**).

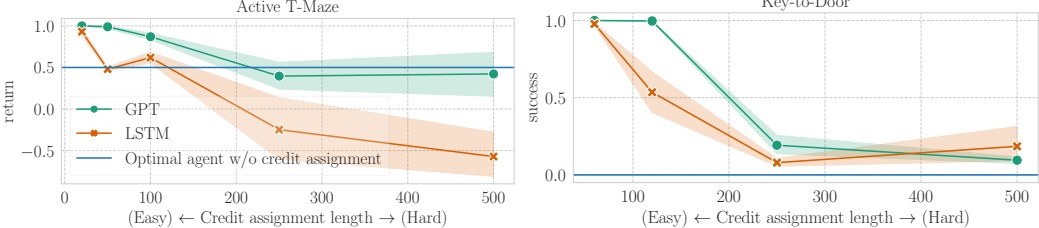

Figure 6: **Transformer-based RL improves temporal credit assignment compared to LSTM-based RL, but its advantage diminishes in long-term scenarios. Left:** results in Active T-Mazes with varying credit assignment lengths from 20 to 500; **Right:** results in Key-to-Door with varying credit assignment lengths from 60 to 500. We also show the performance of the optimal policies that lack (long-term) credit assignment – taking greedy actions to maximize *immediate* rewards. Each data point in the figure represents the final performance of an agent in a task with the credit assignment length indicated on the x-axis, averaged over 10 seeds.

based agents struggle to solve the task with a credit assignment length of 250, as indicated by the low success rates in Fig. 6 (right). Nevertheless, we observe that Transformers can improve credit assignment, especially when the length is medium around 100 in both tasks, compared to LSTMs.

On the other hand, we investigate the performance of multi-layer Transformers on credit assignment tasks to explore the potential scaling law in RL, as discovered in supervised learning (Kaplan et al., 2020). Fig. 7 reveals that multi-layer Transformers greatly enhance performance in tasks with credit assignment lengths up to 100 in Active T-Maze and 120 in Key-to-Door. Yet they fail to improve long-term credit assignment. Interestingly, 4-layer Transformers have similar or worse performance compared to 2-layer Transformers, suggesting that the scaling law might not be seamlessly applied to credit assignment in RL.

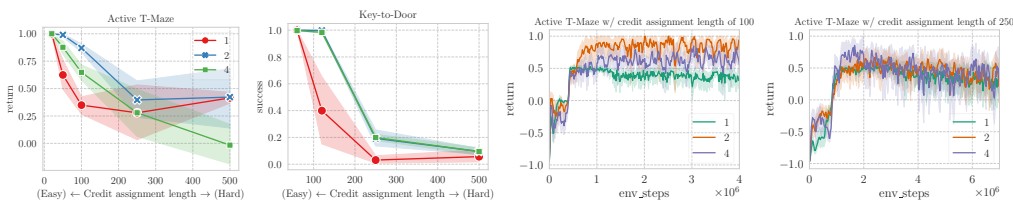

Figure 7: **Increasing the number of layers (and heads) in Transformers aids in temporal, but not long-term, credit assignment.** The left two figures show the final results from Active T-Maze and Key-to-Door, varying the number of layers and heads in Transformers from 1, 2 to 4. The right two figures show the associated learning curves in Active T-Maze with different credit assignment lengths.

## 5.3 Transformers in RL Raise Sample Complexity in Certain Short-Term Memory Tasks

Lastly, we revisit the standard POMDP benchmark used in prior works (Ni et al., 2022; Han et al., 2020; Meng et al., 2021). This PyBullet benchmark consists of tasks where only the positions (referred to as "-P" tasks) or velocities (referred to as "-V" tasks) of the agents are revealed, with dense rewards. As a result, they necessitate short-term credit assignment. Further, the missing velocities can be readily inferred from two successive positions, and the missing positions can be

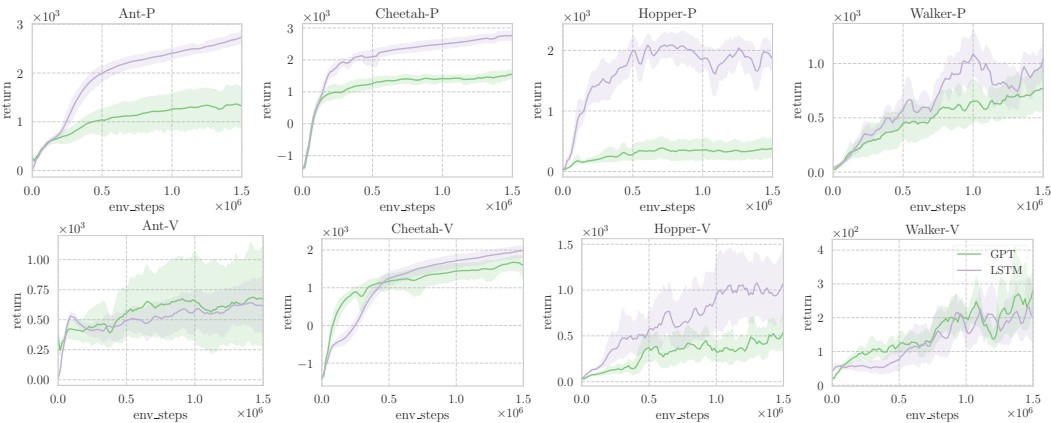

Figure 8: **Transformer-based RL is sample-inefficient compared to LSTM-based RL in most PyBullet occlusion tasks.** The top row shows tasks with occluded velocities, while the bottom row shows tasks with occluded positions. Results are averaged over 10 seeds.

approximately deduced by recent velocities, thereby requiring short-term memory. Adhering to the implementation of recurrent TD3 (Ni et al., 2022) and substituting LSTMs with Transformers, we investigate their sample efficiency over 1.5M steps. We find that in most tasks, Transformer-based agents show worse sample efficiency than LSTM-based agents, as displayed in Fig. 8. This aligns with the findings in POPGym paper (Morad et al., 2023) where GRU-based PPO is more sample-efficient than Transformer-based PPO across POPGym benchmark that requires relatively short memory lengths. These observations are also consistent with the known tendency of Transformers to thrive in situations with large datasets in supervised learning.

# 6 Limitations

We provide quantitative definitions of memory and credit assignment lengths, but developing programs capable of calculating them for any given task remains an open challenge. This can often be desirable since sequence models can be expensive to train in RL agents. The worst compute complexity of $c^{\mathcal{M}}$ can be $O(|\mathcal{S}||\mathcal{A}|T^2)$ for a finite MDP, while the complexity of $m^{\mathcal{M}}$ can be $O((|\mathcal{O}|+ |\mathcal{A}|)^T|\mathcal{O}||\mathcal{A}|T)$ for a finite POMDP. In addition, $c^{\mathcal{M}}$ does not take the variance of intermediate rewards into account. These reward noises have been identified to complicate the learning process for credit assignment (Mesnard et al., 2021). On the other hand, $m^{\mathcal{M}}$ does not take into account the memory capacity (*i.e.*, the number of bits) and robustness to observation noise (Beck et al., 2019).

On the empirical side, we evaluate Transformers on a specific architecture (GPT-2) with a relatively small size in the context of online model-free RL. As a future research direction, it would be interesting to explore how various sequence architectures and history abstraction methods may impact and potentially improve long-term memory and/or credit assignment.

# 7 Conclusion

In this study, we evaluate the memory and credit assignment capabilities of memory-based RL agents, with a focus on Transformer-based RL. While Transformer-based agents excel in tasks requiring long-term memory, they do not improve long-term credit assignment and generally have poor sample efficiency. Furthermore, we highlighted that many existing RL tasks, even those designed to evaluate (long-term) memory or credit assignment, often intermingle both capabilities or require only short-term dependencies. While Transformers are powerful tools in RL, especially for long-term memory tasks, they are not a universal solution to all RL challenges. Our results underscore the ongoing need for careful task design and the continued advancement of core RL algorithms.

## Acknowledgements

This research was enabled by the computational resources provided by the Calcul Québec (`calculquebec.ca`) and the Digital Research Alliance of Canada (`alliancecan.ca`). We thank Pierluca D'Oro and Zhixuan Lin for their technical help. We thank Jurgis Pasukonis and Zhixuan Lin for correcting some factual errors. We also thank anonymous reviewers, Pierluca D'Oro, Siddhant Sahu, Giancarlo Kerg, Guillaume Lajoie, and Ryan D'Orazio for constructive discussion.

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

# Appendix

## Table of Contents

## A  Proofs

**Lemma 1** (**Upper bound of value memory length**).  *For any policy $\pi$,*

$$l_{\text{value}}^{\mathcal{M}}(\pi) \leq \max(m_{\text{reward}}^{\mathcal{M}}, m_{\text{transit}}^{\mathcal{M}}, l_{\text{mem}}(\pi) - 1) \tag{3}$$

*Proof of Lemma 1.*  We show this by expanding the sum of rewards in $Q$-value function:

$$Q^\pi(h_{1:t}, a_t) = \mathbb{E}_{\pi, \mathcal{M}}\left[ \sum_{i=t}^{T} \gamma^{i-t} r_i \mid h_{1:t}, a_t \right] \tag{4}$$

$$= \underbrace{\mathbb{E}[r_t \mid h_{1:t}, a_t]}_{\text{reward memory}} + \sum_{i=t}^{T-1} \gamma^{i+1-t} \tag{5}$$

$$\int \underbrace{P(o_{i+1} \mid h_{1:i}, a_i)}_{\text{transition memory}} \underbrace{\pi(a_{i+1} \mid h_{1:i+1})}_{\text{policy memory}} \underbrace{\mathbb{E}[r_{i+1} \mid h_{1:i+1}, a_{i+1}]}_{\text{reward memory}} do_{t+1:i+1} da_{t+1:i+1} \tag{6}$$

$$\overset{(A)}{=} \mathbb{E}\left[ r_t \mid h_{t-m_{\text{reward}}^{\mathcal{M}}+1:t}, a_t \right] + \sum_{i=t}^{T-1} \gamma^{i+1-t} \tag{7}$$

$$\int P(o_{i+1} \mid h_{i-m_{\text{transit}}^{\mathcal{M}}+1:i}, a_i) \pi(a_{i+1} \mid h_{i-l_{\text{mem}}(\pi)+2:i+1}) \mathbb{E}\left[ r_{i+1} \mid h_{i-m_{\text{reward}}^{\mathcal{M}}+2:i+1}, a_{i+1} \right] do_{t+1:i+1} da_{t+1:i+1} \tag{8}$$

$$= \mathbb{E}_{\pi, \mathcal{M}}\left[ \sum_{i=t}^{T} \gamma^{i-t} r_i \mid h_{\min(t-m_{\text{reward}}^{\mathcal{M}}+1, t-m_{\text{transit}}^{\mathcal{M}}+1, t-l_{\text{mem}}(\pi)+2, t-m_{\text{reward}}^{\mathcal{M}}+2):t}, a_t \right] \tag{9}$$

$$= \mathbb{E}_{\pi, \mathcal{M}}\left[ \sum_{i=t}^{T} \gamma^{i-t} r_i \mid h_{t+1-\max(m_{\text{reward}}^{\mathcal{M}}, m_{\text{transit}}^{\mathcal{M}}, l_{\text{mem}}(\pi)-1):t}, a_t \right] \tag{10}$$

where line $(A)$ follows from the definition of memory lengths in policy (Def. 1.C), reward (Def. 1.A), and transition (Def. 1.B). By the definition of value memory length (Def. 1.D), the shortest history to match the value, we have the upper bound: $l_{\text{value}}^{\mathcal{M}}(\pi) \leq \max(m_{\text{reward}}^{\mathcal{M}}, m_{\text{transit}}^{\mathcal{M}}, l_{\text{mem}}(\pi) - 1)$.

$\square$

**Lemma 2 (Lower bound of value memory length).** *For an optimal policy* $\pi^* \in \operatorname{argmin}_{\pi \in \Pi_{\mathcal{M}}^*}\{l_{\text{mem}}(\pi)\}$, *we have* $l_{\text{mem}}(\pi^*) \leq l_{\text{value}}^{\mathcal{M}}(\pi^*)$.

*Proof.* Proof by contradiction. Suppose $l_{\text{value}}^{\mathcal{M}}(\pi^*) < l_{\text{mem}}(\pi^*)$. We can construct a deterministic policy $\pi'$ that has context length of $l_{\text{value}}^{\mathcal{M}}(\pi^*)$:

$$\pi'(h_{t-l_{\text{value}}^{\mathcal{M}}(\pi^*)+1:t}) = \operatorname*{argmax}_{a_t} Q_{l_{\text{value}}^{\mathcal{M}}(\pi^*)}^{\pi^*}(h_{t-l_{\text{value}}^{\mathcal{M}}(\pi^*)+1:t}, a_t) \tag{11}$$

$$= \operatorname*{argmax}_{a_t} Q^{\pi^*}(h_{1:t}, a_t), \quad \forall t, h_{1:t} \tag{12}$$

This is a greedy policy derived from $\pi^*$'s value function. The policy memory length of $\pi'$ is **strictly shorter** than $l_{\text{mem}}(\pi^*)$, by definition and assumption:

$$l_{\text{mem}}(\pi') \leq l_{\text{ctx}}(\pi') = l_{\text{value}}^{\mathcal{M}}(\pi^*) < l_{\text{mem}}(\pi^*) \tag{13}$$

Similar to policy improvement theorem (Sutton and Barto, 2018, Chap 4.2) in Markov policies, we can show that $\pi'$ is at least as good as $\pi^*$, thus contradicting the assumption that $\pi^*$ is an optimal policy that has the shortest policy memory length. To see this, for any $t$ and $h_{1:t}$,

$$V^{\pi^*}(h_{1:t}) \leq \max_{a_t} Q^{\pi^*}(h_{1:t}, a_t) \tag{14}$$

$$\overset{(A)}{=} \max_{a_t} Q^{\pi^*}(h_{1:t}, \pi'(h_{t-l_{\text{value}}^{\mathcal{M}}(\pi^*)+1:t})) \tag{15}$$

$$\overset{(B)}{=} \mathbb{E}_{\mathcal{M}}\left[r_t + \gamma Q^{\pi^*}(h_{1:t+1}, a_{t+1}) \mid h_{1:t}, a_t \sim \pi', a_{t+1} \sim \pi^*\right] \tag{16}$$

$$\overset{(C)}{\leq} \mathbb{E}_{\mathcal{M}}\left[r_t + \gamma Q^{\pi^*}(h_{1:t+1}, a_{t+1}) \mid h_{1:t}, a_{t:t+1} \sim \pi'\right] \tag{17}$$

$$\cdots \tag{18}$$

$$\leq \mathbb{E}_{\mathcal{M}}\left[\sum_{i=t}^{T} \gamma^{i-t} r_i \mid h_{1:t}, a_{t:T-1} \sim \pi'\right] \tag{19}$$

$$= \mathbb{E}_{\pi',\mathcal{M}}\left[\sum_{i=t} \gamma^{i-t} r_i \mid h_{1:t}\right] = V^{\pi'}(h_{1:t}) \tag{20}$$

where $(A)$ follows the definition of $\pi'$, $(B)$ uses the Bellman equation and $(C)$ uses the property of greedy policy $\pi'$. $\square$

*Proof of Theorem 1.* By Lemma 2, $l_{\text{value}}^{\mathcal{M}}(\pi^*) \geq l_{\text{mem}}(\pi^*)$. On the other hand, by Lemma 1, $l_{\text{value}}^{\mathcal{M}}(\pi^*) \leq \max(m_{\text{reward}}^{\mathcal{M}}, m_{\text{transit}}^{\mathcal{M}}, l_{\text{mem}}(\pi^*) - 1)$. Thus,

$$l_{\text{mem}}(\pi^*) \leq \max(m_{\text{reward}}^{\mathcal{M}}, m_{\text{transit}}^{\mathcal{M}}, l_{\text{mem}}(\pi^*) - 1) \tag{21}$$

This implies

$$l_{\text{mem}}(\pi^*) - 1 < \max(m_{\text{reward}}^{\mathcal{M}}, m_{\text{transit}}^{\mathcal{M}}) \tag{22}$$

$$l_{\text{value}}^{\mathcal{M}}(\pi^*) \leq \max(m_{\text{reward}}^{\mathcal{M}}, m_{\text{transit}}^{\mathcal{M}}) \tag{23}$$

$\square$

$c(\pi^*)$ *may be not unique among all optimal policies.* For example, consider an MDP described in Fig. 9, and two optimal deterministic Markov policy, $\pi_1^*, \pi_2^*$, only different at the state P2: $\pi_1^*(\text{P2}) = x$ while $\pi_2^*(\text{P2}) = y$. It is easy to compute the credit assignment length in this MDP, because only state P1 contains suboptimal action. At state P1, the optimal action $y$ starts to be better than suboptimal action $x$ at the next 2 steps for $\pi_2^*$, while at the next 3 steps for $\pi_1^*$. Thus, $c(\pi_1^*) = 3$ and $c(\pi_2^*) = 2$.

$\square$

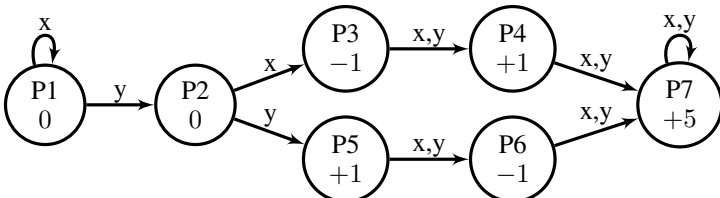

Figure 9: An MDP that shows the credit assignment length over optimal policies is not a constant. Each node is a state with its name on the top and its reward at the bottom. The initial state is P1. The action space is $\{x, y\}$. The finite horizon $T \geq 4$ and $\gamma = 1$.

# B  Details on Environments Related to Memory and Credit Assignment

## B.1  Abstract Problems

**Example 3** ($n$-order MDPs). *It is a POMDP where the recent $n$ observations form the current state, i.e. the transition is $P(o_{t+1} \mid o_{t-n+1:t}, a_t)$ and the reward is $R_t(o_{t-n+1:t}, a_t)$.*

**Example 4** (Episodic reward). *It is a POMDP with a finite horizon of $T$. The reward function is episodic in that $R_t = 0$ for $\forall t < T$.*

**Example 5** (Decomposed episodic reward (Ren et al., 2022)). *It has an episodic reward (Eg. 4), decomposed into the sum of Markovian rewards: $R_T(h_{1:T}, a_T) = \sum_{t=1}^{T} r'(o_t, a_t)$, where $r' : \mathcal{O} \times \mathcal{A} \to \mathbb{R}$ is a Markovian reward function. The transition $P(o_{t+1} \mid o_t, a_t)$ is Markovian. The smallest policy memory length for an optimal policy $l_{\mathrm{mem}}(\pi^*)$ can be shown as 1 (see below).*

*Proof of decomposed episodic reward problem requiring $l_{\mathrm{mem}}(\pi^*) = 1$ at most.* Suppose the policy is non-stationary, composed of $\{\pi_t\}_{t=1}^{T}$. For any time step $t \leq T$ in the POMDP $\mathcal{M}$,

$$Q^{\pi}(h_{1:t}, a_t) = \mathbb{E}_{\pi, \mathcal{M}} \left[ \sum_{k=t}^{T} r_k \mid h_{1:t}, a_t \right] \tag{24}$$

$$= \mathbb{E}_{\pi, \mathcal{M}}[r_T \mid h_{1:t}, a_t] = \mathbb{E}_{\pi, \mathcal{M}} \left[ \sum_{k=1}^{T} r'(o_k, a_k) \mid h_{1:t}, a_t \right] \tag{25}$$

$$= \sum_{k=1}^{t} r'(o_k, a_k) + \mathbb{E}_{\pi_{t+1:T}, o_{t+1} \sim P(\mid o_t, a_t), \mathcal{M}} \left[ \sum_{k=t+1}^{T} r'(o_k, a_k) \mid h_{1:t}, a_t \right] \tag{26}$$

By taking argmax,

$$\operatorname*{argmax}_{a_t} Q^{\pi}(h_{1:t}, a_t) = \operatorname*{argmax}_{a_t} r'(o_t, a_t) + \mathbb{E}_{\pi_{t+1:T}, o_{t+1} \sim P(\mid o_t, a_t), \mathcal{M}} \left[ \sum_{k=t+1}^{T} r'(o_k, a_k) \mid h_{1:t}, a_t \right] \tag{27}$$

Now we show that there exists an optimal policy that is Markovian by backward induction. When $t = T$, (27) reduces to

$$\operatorname*{argmax}_{a_T} Q^{\pi}(h_{1:T}, a_T) = \operatorname*{argmax}_{a_T} r'(o_T, a_T) \tag{28}$$

thus Markovian policy $\pi_T(o_T) = \operatorname{argmax}_{a_T} r'(o_T, a_T)$ is optimal. Now assume $\pi_{t+1:T}$ are Markovian and optimal, consider the step $t$ in (27), all future observations and actions $(o_{t+1:T}, a_{t+1:T})$ rely on $(o_t, a_t)$ by the property of Markovian transition and induction, thus optimal action $a_t^*$ can only rely on $o_t$.

$\square$

**Example 6** (Delayed rewards of $n$ steps (Arjona-Medina et al., 2019)). *The transition $P(o_{t+1} \mid o_t, a_t)$ is Markovian, and the reward $R_t(o_{t-n+1}, a_{t-n+1})$ depends on the observation and action $n$ steps before. Decomposed episodic rewards (Eg. 5) can be viewed as an extreme case of delayed rewards.*

**Example 7** (Delayed execution of $n$ steps (Derman et al., 2021)). *The transition is $P(o_{t+1} \mid o_t, a_{t-n})$, and the reward is $R_t(o_t, a_{t-n})$.*

Table 2: **The memory and credit assignment lengths required in the related abstract problems, using our notion.** Assume all tasks have a horizon of $T$.

| Abstract Problem $\mathcal{M}$ | $l_{\text{mem}}(\pi^*)$ | $m^{\mathcal{M}}_{\text{reward}}$ | $m^{\mathcal{M}}_{\text{transit}}$ | $c^{\mathcal{M}}$ |
|---|---|---|---|---|
| MDP | $[0,1]$ | $[0,1]$ | $[0,1]$ | $[1,T]$ |
| $n$-order MDP (Eg. 3) | $[0,n]$ | $[0,n]$ | $[0,n]$ | $[1,T]$ |
| POMDP | $[0,T]$ | $[0,T]$ | $[0,T]$ | $[1,T]$ |
| Episodic reward (Eg. 4) | $T$ | $T$ | $[0,T]$ | $T$ |
| Decomposed episodic reward (Eg. 5) | $[0,1]$ | $T$ | $[0,1]$ | $T$ |
| Delayed rewards of $n$ steps (Eg. 6) | $[0,1]$ | $n$ | $[0,1]$ | $[n,T]$ |
| Delayed execution of $n$ steps (Eg. 7) | $n$ | $n$ | $n$ | $[n,T]$ |
| Delayed observation of $n$ steps (Eg. 8) | $n$ | $n$ | $n$ | $[0,T]$ |

**Example 8** (Delayed observation of $n$ steps (Katsikopoulos and Engelbrecht, 2003)). *The transition is $P(o_{t+1} \mid o_t, a_{t-n})$, and the reward is $R_t(o_t, a_{t-n:t})$.*

We summarize our analysis in Table 2.

## B.2 Concrete Benchmarks for Memory

**T-Maze (Bakker, 2001).** The task is almost the same as our Passive T-Maze, except that the agent is not penalized if it moves left instead of right in the corridor. The modification results in a credit assignment length closer to $T$. The authors test RL algorithms with the largest horizon of 70.

**MiniGrid-Memory (Chevalier-Boisvert et al., 2018).** This task has the same structure as our Active T-Maze. In the worst-case scenario, the optimal policy can traverse the horizontal corridor back and forth, and then follow the vertical corridor to solve the task. Thus, the memory and credit assignment lengths of optimal policy is upper bounded by three times the maze side length, which is $3 \times 17 = 51$ for a $17 \times 17$ maze (`MiniGrid-MemoryS17Random-v0`). The reward memory length can equal the horizon of 1445.

**Passive Visual Match (Hung et al., 2018).** The environment consists of a partially observable $7 \times 11$ grid-world, where the agent can only observe a $5 \times 5$ grid surrounding itself. The task is separated into three phases. In the first, the agent passively observes a randomly generated color. In the second, the agent must pick up apples with immediate reward. In the final phase, three random colored squares are displayed, and the agent must pick up the one corresponding to the color observed in the first phase, which tests the agent's ability to recall temporally distant events. Since rewards are relatively immediate with respect to the agent's actions, the credit assignment length of optimal policy is upper bounded by the length of the shortest path needed to traverse the grid-world, *i.e.* $7 + 11 = 18$.

**TMaze Long and TMaze Long Noise (Beck et al., 2019).** Both tasks are structurally similar to our Passive T-Maze. In TMaze Long, the agent is forced to take the forward action until reaching the junction (*i.e.*, the action space is a singleton except at the junction). At the junction, the agent is required to choose the goal candidate matching the color it observed at the starting position. Therefore, the memory length is the horizon $T$ and the credit assignment length is 1 since only the terminal action can be credited. TMaze Long Noise introduces uniform noise appended to the observation space. Both tasks have a horizon and corridor length of 100.

**PsychLab (Fortunato et al., 2019).** This environment simulates a lab environment in first person, and can be sub-categorized into four distinct tasks. In all tasks, one or many images are shown, and the overall objective of all tasks is to correctly memorize the contents of the images for future actions. Images are passively observed by the agent, thus eliminating the need for long-term credit assignment, and instead focusing on long-term memory.

**Spot the Difference (Fortunato et al., 2019).** In this task, two nearly identical rooms are randomly generated, and the agent must correctly identify the differences between the rooms. The agent must navigate through a corridor of configurable length to go from one room to another. While this task was designed to test an agent's memory, effective temporal credit assignment is also necessary for an optimal policy. Similar in flavor to **Active Visual Match (Hung et al., 2018)**, an agent must first explore the initial room to be able to identify the differences in the second room. Rewards for this exploratory phase are only given near the end of the task once the agent has identified all differences. Long-term credit assignment is needed to incorporate these actions into an optimal policy.

**Goal Navigation (Fortunato et al., 2019).** A random maze and goal state is generated at the start of each episode. The agent is rewarded every time it reaches the goal state, and is randomly re-initialized in the maze each time it does so. Crucial information about the maze needs to be memorized throughout the episode for optimal performance, while exploratory actions may also be taken early for better quicker navigation later. Consequently, this environment fails to cleanly disentangle temporal credit assignment from memory.

**Numpad (Parisotto et al., 2020; Humplik et al., 2019).** This is a visual RL problem with continuous control. In this task, there is an $N \times N$ number pad, and the goal of the agent is to activate up to $N^2$ pads in a specific order. The order is randomized at the beginning of each episode. The agent does not know the order, and has to explore to figure it out. The agent will receive an immediate reward of $1$ if it hits the next pad correctly; otherwise, the agent must restart. As a result, the credit assignment length is short, although the exact value is hard to determine. The optimal policy needs to memorize the order of $N^2$ pads during the whole episode, thus the memory length is upper bounded by the full horizon of $500$.

**Memory Length (Osband et al., 2020, A.6).** The task is similar to our Passive T-Maze in terms of reward memory and credit assignment lengths. However, we did not adopt this task because the agent cannot change the observation through its action, making the task solvable by supervised learning. Moreover, in this task, the observation is i.i.d. sampled at each time step, making a transition memory length of only $1$. The authors set the horizon at a maximum of $100$.

**Reacher-pomdp (Yang and Nguyen, 2021).** In this task, the goal position in Reacher is only revealed at the first step. The optimal policy has to memorize the goal until it is reached. The policy memory length is thus long, but not necessarily the full horizon. The task provides dense rewards, so the credit assignment length is short, although the exact value is unclear. The horizon is set to $50$.

**T-Maze (Lambrechts et al., 2021).** This task has the same structure as T-Maze (Bakker, 2001). The RL algorithms are tested with the largest horizon of $200$.

**Ballet (Lampinen et al., 2021).** Ballet is a 2D gridworld task with the agent situated at the center. In its most challenging variant, there are 8 dancers around the agent, each executing a 16-step dance sequentially. A 48-step interval separates two consecutive dances. Consequently, the agent observes the environment for a minimum of $16 * 8 + 48 * (8 - 1) = 464$ steps. Following the final dance, the agent receives a reward upon reaching the dancer performing a specific routine. Hence, the optimal policy's memory length is at least $464$ steps, and the reward memory length is capped by the horizon of $1024$. The agent's actions are constrained to the post-dance phase, resulting in a short credit assignment length determined by the time taken to reach the correct dancer.

**HeavenHell (Esslinger et al., 2022; Thrun, 1999).** This task has the same structure as our Active T-Maze. The agent has to first move south to reach the oracle and then move north to reach the heaven. The reward is given only at the terminal step. The credit assignment length is the full horizon, which is around $20$. The Gym-Gridverse and Car Flag tasks in their work have the same structures as HeavenHell but feature more complex observation spaces.

**Memory Cards (Esslinger et al., 2022).** The task can be viewed as a discrete version of Numpad. There are $N$ pairs of cards, with each pair sharing the same value. The agent can observe a random card's value at each step, after which the card value is hidden. The agent then takes action to choose the paired card that matches the observed card. It will get an immediate reward of $0$ if it makes the correct choice, otherwise, a reward of $-1$. The credit assignment length is thus $1$. The minimal memory length is around $O(N)$ to observe all cards' values, but hard to determine exactly due to randomness. The finite horizon is set to $50$.

**Memory Maze (Pasukonis et al., 2022).** In this environment, the agent is placed in a randomly generated $N \times N$ maze with $N = 15$ at most. There are $K$ colored balls in the maze, and the agent must pick up the $K$ colored balls in a specified order (randomized in every episode). As for the reward memory, the reward is given only when the agent picks up the correct ball and solely depends on current observation and action, thus $m_{\text{reward}}^{\mathcal{M}} = 1$. Although their longest horizon $T$ can be $4000$, the credit assignment length is upper bounded by the size of the maze. In the worst-case scenario, an optimal policy can perform a naive search on the entire maze and uncover the maze plan to obtain a reward, which costs $O(N^2)$ steps. Nevertheless, the optimal policy memory length can be much longer than $O(N^2)$ steps and close to the horizon $T$. This is because the optimal policy may need to memorize the initial subsequence of observations to infer the maze plan when taking current actions.

**POPGym** (Morad et al., 2023). This benchmark comprises 15 tasks from which we examine several representative ones:

- The diagnostic POMDP task, Repeat First, provides a reward based on whether the action repeats the first observation. Thus, the credit assignment length is 1. Although this task indeed has a reward memory length of $T$, the optimal policy memory length can be as low as 2 simply by recalling the previous optimal action. The horizon $T$ of this task can be 16 decks (831 steps).

- In another diagnostic POMDP task, Autoencode, an agent first observes a sequence of cards, and then must reproduce the exact sequence in a reverse order to earn rewards. Here, the credit assignment length is 1, and the optimal policy memory length is at most twice the number of cards, which is 6 decks (311 steps).

- The control POMDP tasks, Stateless Cartpole and Pendulum, occlude velocities, making them similar to the "-P" tasks we previously evaluated. These tasks require short-term dependencies.

- The game POMDP task, Battleship, requires the agent to deduce the locations of all ships on a board without observing the board. This requires a memory length upper bounded by the area of the board, *i.e.* $12^2 = 144$, and short-term credit assignment. Similarly, the Concentration game requires the agent to maximize the number of card matches in an episode. Although the reward memory length of horizon $T$, the optimal policy memory length can be restricted to the number of cards, *i.e.* $2 * 52 = 104$.

Overall, this benchmark indeed emphasizes the evaluation of memory capabilities with short-term credit assignment, where the memory length is limited by the maximal horizon of around 831.

**Memory Gym** (Pleines et al., 2023). This benchmark has a pixel-based observation space and a discrete action space for controlling movement within a gridworld. The Mortar Mayhem (MM) task is considered as one of the hardest tasks in this benchmark. MM requires the agent to first memorize a sequence of five commands, and then execute the exact commands in order. A reward is given when the agent successfully executes one command. Thus, the credit assignment length for optimal policy is at most the area of the gridworld, which is $5 * 5 = 25$. The memory length is at most the full horizon, which according to their Table 3 is at most 135.

### B.3 Concrete Benchmarks for Credit Assignment

**Active Visual Match** (Hung et al., 2018). The agent has to open the door to observe the color in the first phase, picks apples up in the second phase, and goes to the door with the corresponding color in the final phase. Thus, the policy needs to memorize the color.

**Episodic MuJoCo** (Liu et al., 2019; Ren et al., 2022). It is a decomposed episodic reward problem (Eg. 5). The terminal reward is the sum of the original Markov dense rewards. They tried the maximal horizon as 1000.

**Umbrella Length** (Osband et al., 2020, A.5). It is an MDP and the initial action taken by the agent will be carried through the episode. The terminal reward depends on the terminal state, which is determined by the initial action. Thus, the credit assignment length is the full horizon, at most 100. There are also some random intermediate rewards that are independent of actions.

**Key-to-Door** (Raposo et al., 2021; Mesnard et al., 2021; Chen et al., 2021). The agent has to reach the key in the first phase, picks apples up in the second phase, and goes to the single door. Thus, the policy has little memory, while the terminal reward depends on whether the agent picks the key up in the first phase. In this sense, it can be also viewed as a decomposed episodic reward problem with some noisy immediate rewards. Raposo et al. (2021) tried the maximal horizon as 90. The horizon in Mesnard et al. (2021); Chen et al. (2021) is unknown but should be similar.

**Catch with delayed rewards** (Raposo et al., 2021). It is a decomposed episodic reward problem (Eg. 5). The terminal reward is given by how many times the agent catches the ball in an episode. They tried the maximal horizon as 280.

**Push-r-bump** (Yang and Nguyen, 2021). The agent has to first explore to find out the correct bump, and then move towards it to receive a terminal reward. The credit assignment length is thus long. The horizon is 50.

## C Experiment Details

### C.1 Memory-Based RL Implementation

Our implementation builds upon prior work (Ni et al., 2022), providing a strong baseline on a variety of POMDP tasks, including Key-to-Door and PyBullet tasks used in this work. We denote this implementation[9] as `POMDP-baselines`. All parameters of our agents in our experiments are trained end-to-end with model-free RL algorithms from scratch. The following paragraphs provide detailed descriptions.

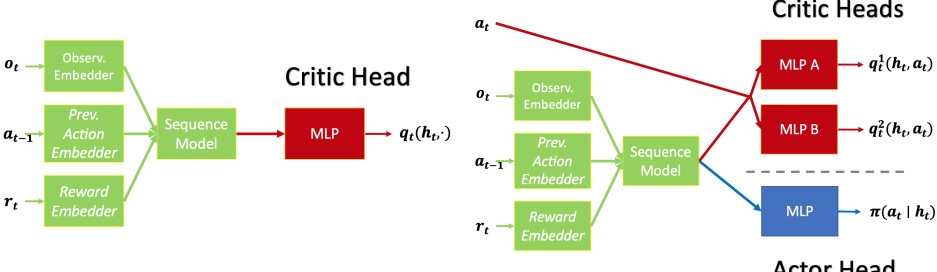

Figure 10: **Agent architectures** for memory-based DDQN (left) and TD3 (right) in our implementation. While rewards are not used as inputs in our experiments, they are included in this figure for completeness. Compared to `POMDP-baselines` (Ni et al., 2022), we have removed shortcuts from $o_t$ to the MLPs and shared the same sequence model between the actor and critic.

**Sharing the sequence model in actor-critic with frozen critic parameters in the actor loss.** Our agent architecture, depicted in Fig. 10, simplifies the one in `POMDP-baselines` by sharing the sequence model between the actor and critic. Previously, `POMDP-baselines` found sharing the RNNs between actor and critic causes instability in gradients, thus they adopt the separate sequence encoders for actor and critic. We examine their code and find that it is mainly due to the gradient bias in the actor loss. This issue is also discussed in prior (Yang and Nguyen, 2021) and concurrent works (Grigsby et al., 2023) on memory-based RL and related to sharing CNN encoders in visual RL (Yarats et al., 2021a,b). Below we formalize the issue and introduce our solution.

In the **separate encoder** setting, we consider an actor and a critic with distinct history encoders $f_{\phi_\pi}$ and $f_{\phi_Q}$, mapping a history $h$ into latent states $z_\pi$ and $z_Q$, respectively. The history encoder parameters are denoted as $\phi_\pi$ and $\phi_Q$. The actor is parameterized by $\pi_\nu(f_{\phi_\pi}(h))$ and the critic is parameterized by $Q_\omega(f_{\phi_Q}(h), a)$ with $\nu$ and $\omega$ denoting their respective MLP parameters. Considering the DDPG algorithm (Lillicrap et al., 2016) for training[10] and a tuple of data $(h, a, o', r)$[11], the loss of critic parameters is

$$L_Q(\phi_Q, \omega) = \frac{1}{2}(Q_\omega(f_{\phi_Q}(h), a) - Q^{\text{tar}}(h, a, o', r))^2, \tag{29}$$

where $Q^{\text{tar}}(h, a, o', r) := r + \gamma Q_{\overline{\omega}}(f_{\overline{\phi_Q}}(h'), \pi_{\overline{\nu}}(f_{\overline{\phi_\pi}}(h')))$ and $\overline{\theta}$ denotes the stopped-gradient version of $\theta$ and $h' = (h, a, o')$. The loss of actor parameters is

$$L_\pi(\phi_\pi, \nu) = -Q_\omega(f_{\phi_Q}(h), \pi_\nu(f_{\phi_\pi}(h))). \tag{30}$$

The total loss of actor and critic parameters is thus $L_{\text{sep}}(\phi_\pi, \phi_Q, \omega, \nu) := L_Q(\phi_Q, \omega) + L_\pi(\phi_\pi, \nu)$. The gradients of $L_{\text{sep}}$ are

$$\nabla_{\phi_Q} L_{\text{sep}} = (Q_\omega(f_{\phi_Q}(h), a) - Q^{\text{tar}}(h, a, o', r))\nabla_{z_Q} Q_\omega(f_{\phi_Q}(h), a)\nabla_{\phi_Q} f_{\phi_Q}(h) \tag{31}$$

$$\nabla_\omega L_{\text{sep}} = (Q_\omega(f_{\phi_Q}(h), a) - Q^{\text{tar}}(h, a, o', r))\nabla_\omega Q_\omega(f_{\phi_Q}(h), a) \tag{32}$$

$$\nabla_{\phi_\pi} L_{\text{sep}} = -\nabla_a Q_\omega(f_{\phi_Q}(h), \pi_\nu(f_{\phi_\pi}(h)))\nabla_{z_\pi} \pi_\nu(f_{\phi_\pi}(h))\nabla_{\phi_\pi} f_{\phi_\pi}(h) \tag{33}$$

$$\nabla_\nu L_{\text{sep}} = -\nabla_a Q_\omega(f_{\phi_Q}(h), \pi_\nu(f_{\phi_\pi}(h)))\nabla_\nu \pi_\nu(f_{\phi_\pi}(h)) \tag{34}$$

---

[9]https://github.com/twni2016/pomdp-baselines

[10]The analysis can be generalized to stochastic actors by the reparametrization trick.

[11]It denotes current history, current action, next observation, and current reward.

Now we consider an actor and a critic with a **shared encoder** denoted as $f_\phi$, mapping a history $h$ into a latent state $z$. The history encoder has parameters $\phi$. The actor is parameterized by $\pi_\nu(f_\phi(h))$ and the critic is parameterized by $Q_\omega(f_\phi(h), a)$. The total loss of actor and critic

$$L_{\text{sha}}(\phi, \omega, \nu) = L_Q(\phi, \omega) - Q_\omega(f_\phi(h), \pi_\nu(f_\phi(h))) \tag{35}$$

has such gradients:

$$
\begin{aligned}
\nabla_\phi L_{\text{sha}} &= (Q_\omega(f_\phi(h), a) - Q^{\text{tar}}(h, a, o', r))\nabla_z Q_\omega(f_\phi(h), a)\nabla_\phi f_\phi(h) \\
&- \nabla_a Q_\omega(f_\phi(h), \pi_\nu(f_\phi(h)))\nabla_z \pi_\nu(f_\phi(h))\nabla_\phi f_\phi(h) - \underbrace{\nabla_z Q_\omega(f_\phi(h), \pi_\nu(f_\phi(h)))\nabla_\phi f_\phi(h)}_{\text{extra gradient term}}
\end{aligned} \tag{36}
$$

$$\nabla_\omega L_{\text{sha}} = (Q_\omega(f_\phi(h), a) - Q^{\text{tar}}(h, a, o', r))\nabla_\omega Q_\omega(f_\phi(h), a) - \underbrace{\nabla_\omega Q_\omega(f_\phi(h), \pi_\nu(f_\phi(h)))}_{\text{extra gradient term}} \tag{37}$$

Comparing the two gradient formulas across the two settings, we find the **extra gradient terms** using the same value of $\phi$ for $\phi_\pi$ and $\phi_Q$ in $L_{\text{sep}}$:

$$
\begin{aligned}
&\nabla_{\phi_\pi} L_{\text{sep}}(\phi, \phi, \omega, \nu) + \nabla_{\phi_Q} L_{\text{sep}}(\phi, \phi, \omega, \nu) - \nabla_\phi L_{\text{sha}}(\phi, \omega, \nu) \\
&= \nabla_z Q_\omega(f_\phi(h), \pi_\nu(f_\phi(h)))\nabla_\phi f_\phi(h)
\end{aligned} \tag{38}
$$

$$\nabla_\omega L_{\text{sep}}(\phi, \phi, \omega, \nu) - \nabla_\omega L_{\text{sha}}(\phi, \omega, \nu) = \nabla_\omega Q_\omega(f_\phi(h), \pi_\nu(f_\phi(h))) \tag{39}$$

To remove these extra gradient terms, we propose freezing the critic parameters $\phi$ and $\omega$ in the actor loss for the shared encoder setting, resulting in a modified loss function $L_{\text{sha-ours}}(\phi, \omega, \nu)$:

$$L_{\text{sha-ours}}(\phi, \omega, \nu) = L_Q(\phi, \omega) - Q_{\overline{\omega}}(f_{\overline{\phi}}(h), \pi_\nu(f_\phi(h))). \tag{40}$$

This offers a theoretical explanation for the practice of detaching critic parameters in the actor loss, adopted in prior works (Yarats et al., 2021a,b; Yang and Nguyen, 2021). It is worth noting that these works take an additional step of freezing the encoder parameters present in actions, resulting in an expression like $Q_{\overline{\omega}}(f_{\overline{\phi}}(h), \pi_\nu(f_{\overline{\phi}}(h)))$. In our preliminary experiments conducted on PyBullet tasks, we observed negligible differences in the empirical performance between their method and ours. Consequently, to maintain consistency with our theoretical insights, we have chosen to implement our approach $L_{\text{sha-ours}}$ in all of our experimental setups.

**Implementation of sequence encoders in our experiments.** We train both LSTM and Transformer sequence encoders with a full context length equal to the episode length on all tasks, except for PyBullet tasks where we follow `POMDP-baselines` to use a context length of $64$. The training input of sequence encoders is based on a history of $t$ observations and $t$ actions (with the first action zero-padded). The history is embedded as follows according to the `POMDP-baselines`. For state-based tasks (T-Mazes and PyBullet), the observations and actions are individually embedded through a linear and ReLU layer. For pixel-based tasks (Passive Visual Match and Key to Door), the observations (images) are embedded through a small convolutional neural network. For all tasks, the sequence of embedded observations and actions are then concatenated along their final dimension to form the input sequence to the sequence model.

For **LSTMs** (Hochreiter and Schmidhuber, 1997), we train LSTMs with a hidden size of $128$, varying the number of layers from $1, 2, 4$. We find single-layer LSTM performs best in T-Mazes and Passive Visual Match, and two-layer best in Key-to-Door, which are reported in Fig. 3 and Fig. 6. See Table 3 for details.

For **Transformers**, we utilize the GPT-2 model (Radford et al., 2019) implemented by Hugging Face Transformer library (Wolf et al., 2019). Our Transformer is a stack of $N$ layers with $H$-headed self-attention modules. It includes causal masks to condition only on past context. A sinusoidal positional encoding (Vaswani et al., 2017) is added to the embedded sequence. The same dropout rate is used for regularization on the embedding, residual, and self-attention layers. We tune $(N, H)$ from $(1, 1), (2, 2), (4, 4)$. We find $(N, H) = (1, 1)$ performs best in Passive T-Maze and Passive Visual Match, and $(2, 2)$ best in Active T-Maze and Key-to-Door, which are reported in Fig. 3 and Fig. 6. See Table 4 for details.

For PyBullet tasks, we follow `POMDP-baselines` to use single-layer LSTMs and Transformers.

Table 3: **LSTM** hyperparameters used for all experiments.

| | Hyperparameter | Value |
|---|---|---|
| State embedder | Obs. embedding size | 32 |
| | Act. embedding size | 16 |
| Pixel embedder | No. channels | (8, 16) |
| | Kernel size | 2 |
| | Stride | 1 |
| | Obs. embedding size | 100 |
| | Act. embedding size | 0 |
| LSTM | Hidden size | 128 |
| | No. layers | 1, 2, 4 |

Table 4: **Transformer** hyperparameters used for all experiments.

| | Hyperparameter | Value |
|---|---|---|
| State embedder | Obs. embedding size | 64 |
| | Act. embedding size | 64 |
| Pixel embedder | No. channels | (8, 16) |
| | Kernel size | 2 |
| | Stride | 1 |
| | Obs. embedding size | 100 |
| | Act. embedding size | 0 |
| GPT | Dropout | 0.1 |
| | No. heads ($H$) | 1, 2, 4 |
| | No. layers ($N$) | 1, 2, 4 |

**Implementation of RL algorithms in our experiments.** We use SAC-Discrete (Christodoulou, 2019) for pixel-based tasks, and TD3 (Fujimoto et al., 2018) for PyBullet tasks, following `POMDP-baselines`. We use DDQN (van Hasselt et al., 2016) for T-Maze tasks, which outperforms SAC-Discrete in our preliminary experiments. We use epsilon-greedy exploration strategy in DDQN with a linear schedule, where the ending epsilon is $\frac{1}{T}$ with $T$ being the episode length. This ensures that the probability of *always* taking deterministic actions throughout an episode asymptotically approaches a constant:

$$\lim_{T \to +\infty} (1 - \epsilon)^T = \lim_{T \to +\infty} \left( 1 - \frac{1}{T} \right)^T = \frac{1}{e} \approx 0.368 \tag{41}$$

where $e$ is the base of the natural logarithm. This approach is critical to solving T-Maze tasks which strictly require a deterministic policy. Table 5 summarizes the details of RL agents.

Table 5: **RL agent** hyperparameters used in all experiments.

| | Hyperparameter | Value |
|---|---|---|
| | Network hidden size | (256, 256) |
| | Discount factor ($\gamma$) | 0.99 |
| | Target update rate | 0.005 |
| | Replay buffer size | $10^6$ |
| | Learning rate | 0.0003 |
| | Batch size | 64 |
| DDQN | epsilon greedy schedule | linear$(1.0, \frac{1}{T}, $ `schedule_steps`$)$ |
| | `schedule_steps` | $0.1 *$ `num_episodes` |
| SAC-Discrete | entropy temperature | 0.1 |

Table 6: Training hyperparameters in our experiments.

| Tasks | context_length | num_episodes |
|---|---|---|
| Passive T-Maze | 1500 | 8000 |
| Active T-Maze | 500 | 4000 |
| Passive Visual Match | 1000 | 10000 |
| Key-to-Door | 500 | 4000 |

## C.2 Additional Results

Fig. 11 shows the learning curves of training Transformers with varying numbers of layers and heads.

Similar to the scaling experiments on Transformers we demonstrated in Sec. 5.2, we conduct an ablation study on scaling the number of layers in LSTMs from 1, 2 to 4 in credit assignment tasks. Fig. 12 shows the results that multi-layer (stacked) LSTMs do not help performance, aligned with empirical findings in stacked LSTMs (Pascanu et al., 2014).

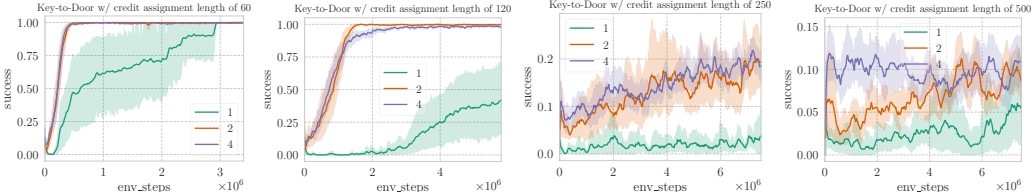

Figure 11: Learning curves of scaling the number of layers and heads in Transformers in Key-to-Door tasks, associated with Fig. 7.

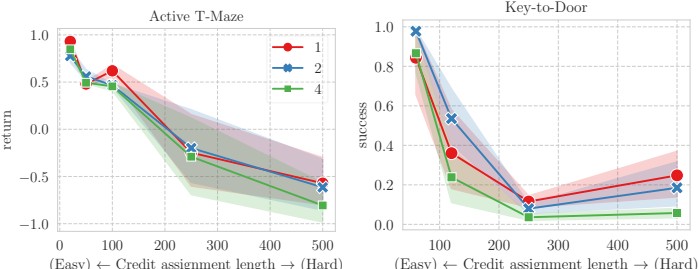

Figure 12: Scaling the number of layers in LSTMs does not affect performance much.

## C.3 Plotting Details

For all the learning curve plots in this paper, we use `seaborn.lineplot` (Waskom, 2021) to show the mean and its $95\%$ confidence interval across 10 seeds. For the aggregation plots, we also use `seaborn.lineplot` to report the final performance, which is averaged over the evaluation results during the last $5\%$ of interactions.

## C.4 Training Compute

Here we report the training compute of the experiments in Sec. 5.1 and Sec. 5.2, because the PyBullet experiments (Sec. 5.3) are relatively cheap to compute. Each experiment run was carried out on a single A100 GPU and a single CPU core. For Transformer-based RL, the GPU memory usage is approximately proportional to the square of the context lengths, with a maximum usage of 4GB for Passive T-Maze with a context length of 1500. For LSTM-based RL, GPU memory usage is roughly linearly proportional.

In our tasks, the context length equals the episode length (and also memory length), thus the total training time is proportional to `num_episodes * context_length**2 * update_frequency` for both Transformers and LSTMs. The `update_frequency`, set as $0.25$, is the ratio of parameter update w.r.t. environment step. Additionally, for multi-layer Transformers, the training time is roughly linear to the number of layers. Table 6 summarizes the training hyperparameters.

In Passive T-Maze with a memory length of 1500, it took around 6 and 4 days to train Transformer-based and LSTM-based RL, respectively. In Passive Visual Match with a memory length of 1000, it took around 5 days for both Transformer-based and LSTM-based RL.

## D Broader Impacts

Our work enhances understanding of long-term memory capability in Transformer-based RL. This improved memory capability, while offering advancements in RL, may pose privacy concerns if misused, potentially enabling systems to retain and misuse sensitive information. As this technology develops, strict data privacy measures are essential. However, negative impacts directly tied to our foundational research are speculative, as we propose no specific application of this technology.

