# OpenReview forum: "When Do Transformers Shine in RL? Decoupling Memory from Credit Assignment"
_NeurIPS.cc/2023/Conference — NeurIPS 2023 oral_

### Official Review · Reviewer_2AdR · 2023-07-04

**Soundness:** 2 fair
**Presentation:** 3 good
**Contribution:** 3 good
**Rating:** 6
**Confidence:** 5

**Summary:**

The paper proposes a set of metrics to measure and isolate the memory dependency of partially observable environments.
The authors illustrate that some currently existing benchmarks do not sufficiently isolate the memory of an agent.
Based on their analysis they propose two versions of a T-Maze environment, one of which appropriately isolates memory.
Further, the authors investigate the effect of memory architecture on tasks that require long- and short-term memory.
Transformer-based policies outperform recurrent policies on tasks that require long-term memory dependencies,
while there seems to be no benefit on tasks that require short-term memory dependencies.

**Strengths:**

**Significance:**

I fully agree with the authors that prior works often conflate different effects and do not sufficiently isolate the memory component.
Therefore metrics that help disentangle memory from other environmental effects are crucial.

**Originality:**

The authors propose novel metrics that evaluate and isolate different aspects of POMDPs in RL, such as credit assignment and long-term memory.
The proposed metrics provide an important measure on the effect of memory and will be useful to facilitate future benchmark design and evaluation of new algorithms.

**Quality:**

Quality of theoretical contributions is high.

**Weaknesses:**

**Scalability:**

I credit the authors for mentioning the complexity for computing c^M as limitation, but it seems that the computation of other metrics is also limited.
For example, l_{mem}(\pi^\asterisk) requires access to the optimal policy.
For the minimalistic TMazes it is straightforward to obtain this policy, how is it obtained for more complex tasks such as Psychlab? Does it involve human experts?
How is it computed for procedurally generated environments, where the optimal policy varies between levels?
How would this scale to more complex environments, given that human demonstrations may not necessarily be optimal?

**Relevance of proposed environments:**

The authors propose two environments, namely Passive T-Maze and Active T-Maze to disentangle credit assignment from memory.
The Minigrid benchmark suite already provides a T-Maze environment that should exhibit the same characteristics as the proposed Active T-Maze, namely MiniGrid-Memory [1].
Passive T-Maze could be useful, but it is very minimalistic and only evaluates whether a small bit of information can be stored and carried across long timespans in an agent's memory.
There is no mention of the observation space, but providing, say, image-based observation for Passive T-Maze, would increase its complexity and enable application of vision-based methods.

**Interpretation of results:**

The authors mention that LSTM starts to falter at a memory length of 250 for Passive T-Maze.
Figure 2, however, shows that LSTM can solve the same environment with a memory length of 750.
This suggests that some other effects might influenced LSTM training, such as hyperparameter tuning.
Maybe the authors can improve the results of LSTM by further tuning of the method?

**Clarity:**

The authors should clearly state that the paper focuses on long-term memory and does not consider other aspects, such as memory capacity, or robustness of memory to noise.

**Missing relevant work:**

There is prior work that derives theoretical bounds on approximation error of history-based methods [2].
Further, other history-based approaches such as a hierarchical Transformer memory [3] and pretrained models as a memory module in RL [4], are not cited.
If applicable, adding these method in their experiments would strengthen the paper and might yield some more interesting findings.

[1] Maxime Chevalier-Boisvert, Lucas Willems, and Suman Pal. Minimalistic Gridworld Environment for OpenAI Gym, 2018. Publication Title: GitHub repository. https://minigrid.farama.org/environments/minigrid/MemoryEnv/

[2] Gandharv Patil et al., On learning history based policies for controlling markov decision processes. 2022.

[3] Andrew Lampinen et al., Towards mental time travel: a hierarchical memory for reinforcement learning agents. NeurIPS 2021

[4] Fabian Paischer et al., History compression via language models in reinforcement learning. ICML 2022


**Questions:**

- The authors conclude that Transformers exhibit worse sample efficiency than LSTM on short-term memory tasks.
Is there an intuition why that would be the case?
On Passive Visual Match, which requires short-term memory according to the authors, a GPT reaches higher return compared to LSTM.
Could it be that the drop in performance is due to different observation/action spaces (continuous vs discrete) instead of long-term vs short-term memory?

- In line 304 the authors claim that the success on long-term memory tasks for GPT2 is consistent with the tendency to perform well in the supervised learning setup on large datasets.
The authors draw an analogy here which lacks support.
How do long-term memory tasks relate to the supervised learning setup on large datasets?
Do the authors imply here, that there are only long-term dependencies present in large datasets used for supervised learning?
Also, Morad et al. 2023 specifically advises against drawing comparisons between supervised learning and RL.

- What is the observation space for the Active/Passive T-Mazes?

- In line 270 it would be good to explain what the actual task is in Passive Visual Match, prior to discussion on the reward function.

- Figure 4 left shows superiority of GPT2 over LSTM, for Passive Visual Match the difference is not as pronounced, a significance test would benefit the interpretation of these results.

- Figure 5: it would be good to clarify what is meant by "optimal policies that lack (long-term) credit assignment"
I assume it refers to a markovian policy, since the return on T-Maze is 0.5.

- Do all figures show mean and standard deviation?
This should be mentioned in the figure captions.

- The last three sentences in Definition 1 could be moved to Definition 2, since they are about memory length and not context length.

- Equation in line 103: I think there should be $t - l_{ctx}(\pi)+1:k$ instead of $t - l_{ctx}(\pi)+1:t$ in the subscript after the conditional independence, right?

- The symbol k is reassigned repeatedly which is a bit confusing, the authors may consider using different symbols.

- Will the authors make the code for reproducing their results and computing their metrics publicly available?


**Limitations:**

The authors have adressed limitations regarding computation of the metrics in more complex environments.
However, it is not clear from the paper, how well the other metrics scale with complexity of the environment, or how the optimal policies are obtained, i.e. does that require human demonstration?
In the case of procedurally generated environments (which are commonly used nowadays) would that require extensive "labeling" by humans?

---

> ### Author Rebuttal · Authors · 2023-08-09
>
>
> > Does computing memory lengths involve human demonstrations?
>
> We here clarify that computing these metrics *does not* require any human demonstrations. For some tasks like T-Maze, we obtain the precise numbers with careful analysis of the task structures. For other complex tasks like Memory Maze and proc-gen environments, as written in L576-579, it's hard for human experts to determine *exact* lengths by problem definition, although some bounds can still be derived, as shown in Table 1.
> We will clarify how we compute the metrics in the camera-ready version.
>
> > Will you release code?
>
> Yes, we have released our code in the supplementary material. We do not use a program to compute our metrics.
>
> > How does computing memory metrics scale with the task?
>
> The complexity depends on the history space, which grows exponentially in a finite POMDP.
> The worst-case complexity for reward and value memory lengths is $O(T (|\mathcal O| + |\mathcal A|)^T |\mathcal A|)$.
> The transition and policy memory lengths' complexity is further multiplied by the cardinality of observation and action spaces.
>
> > The MiniGrid-Memory task already shares the same properties as the proposed Active T-Maze.
>
> Thank you for pointing this out! We will add MiniGrid-Memory task into Table 1. While this task has the same properties as Active T-Maze, it is easier in terms of memory and credit assignment lengths.
> MiniGrid-Memory (sized $13\times 13$) has its memory and credit assignment lengths of optimal policies $\le 3\times 13 = 39$, much smaller than the lengths of 250 in Active T-Maze.
> In addition, MiniGrid-Memory's $147$-dim observations and sparse rewards add unnecessary complexity to evaluation.
> Active T-Maze simplifies the problem with 2-dim observations and dense penalties.
> Lastly, combined with Passive T-Maze, Active T-Maze can be used to evaluate the bottleneck of memory or credit assignment. These advantages motivate us to propose Active T-Maze.
>
>
> > What is the observation space of T-Mazes?
>
> These tasks have 2-dim discrete observations indicating the position of an agent: Oracle, Start, Junction, Goal candidates, Corridors. We have revised the paper to include these details.
>
> > Passive T-Maze is useful but minimalistic. It only evaluates if a small bit can be stored and carried across long timespans in memory.
>
> You are correct, and that was indeed our intention. We chose this minimalistic task to focus on "memory length" rather than "memory capacity" in terms of bits. With no prior evidence that RL agents can maintain a memory spanning $1500$ steps, this task precisely targets this capability. To avoid confusion, we have updated our abstract to state "memorizing observations 1500 steps ago" instead of "memorizing 1500 observations".
>
> > How about trying an image-based version of Passive T-Maze?
>
> In fact, we have conducted experiments on an image-based version of Passive T-Maze, the Passive Visual Match.
>
> > The paper should clarify its focus on long-term memory, not capacity or robustness.
>
> Yes, we will do it.
>
> > Missing related work [2,3,4]. If applicable, running these methods?
>
> Thanks for pointing them out. We find that the hierarchical chunk attention memory [3] is related and will include it into the camera-ready version. It's worth noting that [3] uses observation reconstruction for sparse-reward tasks, unlike our model-free RL. Due to time limit, we cannot run experiments on this method.
> [2] and [4] are remotely related -- [2] focuses on recurrent-based RL  *convergence* and [4] uses pre-trained and frozen language models for RL.
>
>
> > The authors say GPT is less sample-efficient than LSTM in short-term memory tasks, but GPT outperforms LSTM in Passive Visual Match. Could the drop be due to observation/action spaces (continuous vs discrete)?
>
> Yes, it is possible. As stated in L278 and L292, we find that GPTs are sample-efficient on Passive Visual Match, but not PyBullet, though both require short-term memory.
>
> >  Any intuition why GPT is less sample-efficient than LSTM on short-term memory tasks?
>
> Our intuition is that GPTs have fewer inductive bias than LSTMs, thus generally require more data to learn.
>
> > How do long-term memory tasks relate to the SL setup on large datasets?
>
> We did not relate them.
>
> > Do the authors imply that there are only long-term dependencies in large datasets for SL?
>
> No.
>
> > Morad et al. 2023 advises against comparing SL and RL.
>
> We think **purely long-term memory tasks** in RL and SL are related. For example, Copy task [Arjovsky et al., ICML 2016]  is like an SL version of Passive T-Maze. LSTMs failed to solve the long-term Copy task due to gradient training issue, mirroring our findings in Passive T-Maze.
>
> > In L270 it'd be good to explain what the actual task is.
>
> We will add these descriptions in the camera-ready version.
>
> > Fig 5: what is optimal policy that lacks (long-term) credit assignment?
>
> Such a policy maximizes its actions depending on *immediate* rewards only, according to definition of the credit assignment length. In Active T-Maze task, such a policy happens to be Markovian, but it can have memory in general.
> This is a good spot, and we will clarify it in the camera-ready.
>
>
> > Do all figures show mean and std dev? This should be mentioned in captions.
>
> All figures are generated by seaborn.lineplot that shows the mean and its 95\% confidence interval, but not std dev. We will add these descriptions to the captions.
>
> > The last 3 sentences in Def. 1 can be moved to Def. 2, as they are on memory length and not context length.
>
> We will separate the last three sentences from the paragraph to create a new paragraph for Def. 2.
>
> > Question on equation in L103?
>
> Actually, these two are equivalent. For random variables $X,Y,Z$, $X\perp Y \mid Z$ is equivalent to $X \perp Y,Z \mid Z$, where $X = a_t$, $Y = h_{t-l_{ctx}(\pi)+1:t-k}$, and $Z = h_{t-k+1:t}$ in this case.
>
> > The symbol k is reassigned, maybe using other symbols?
>
> We will rename these symbols.

---

> > ### Comment · Reviewer_2AdR · 2023-08-11
> >
> > I greatly appreciate the additional experiments on LSTMs given the limited time window for the rebuttal.
> > Overall the response clarified most of my concerns which is why I decided to increase my rating to 6.

---

### Official Review · Reviewer_kd6F · 2023-07-05

**Soundness:** 3 good
**Presentation:** 4 excellent
**Contribution:** 4 excellent
**Rating:** 7
**Confidence:** 4

**Summary:**

This work explores the impact of memory and credit assignment in decision transformer architectures, presenting several significant claims. Regarding memory, the authors establish an upper bound on the memory length required for an optimal policy. In terms of credit assignment, they provide a lower bound on the number of future steps needed.

Memory length is defined as the number of steps that an action distribution depends on, representing a more recent and shorter dependency than the entire input length. This length, is defined in relation to the value function, transition, and reward.

Credit assignment length of the policy is defined as the number of future steps required for a greedy action to yield a higher discounted sum of rewards compared to a non-greedy action.


**Strengths:**

- This definitions of memory length and credit assignment length appear intuitive and sound.
- Theorem 1 appears clear and mathematically sound.
- A toy environment is presented to illustrate scenarios that are either heavily reliant on memory or credit assignment.  This experiment is intuitive and sound. In the memory task, transformers perform optimally in long memory length tasks, while LSTMs struggle with long-term memory. However, in more complex tasks, the results are not as clear-cut.  These results are significant to the DT community.
- For credit assignment tasks, transformers do not outperform LSTMs and exhibit poor performance at longer credit assignment lengths. They also demonstrate lower sample efficiency compared to LSTMs.  Again, this results is significant to the DT community.
- Overall, the experiments are well-designed, clear, and straightforward, providing a substantial contribution to the field of decision transformer research.

This work is highly novel and is the first to specifically investigate the benefits of transformers in either memory or credit assignment.

**Weaknesses:**

- Can the authors provide any intuition on the failure of transformers and LSTM and transformers in the long term credit assignment tasks? I find it interesting that they both have the same performance trajectory.
- It would be nice to see experimental evaluation of Transformers and LSTM in additional environments other than the toy tasks for better context.

**Questions:**

Above

**Limitations:**

Yes

---

> ### Author Rebuttal · Authors · 2023-08-09
>
> > This work explores the impact of memory and credit assignment in decision transformer architectures.
>
> We would like to clarify that our experiments evaluate the *online model-free RL* setting, not the offline RL setting that is related to Decision Transformers (DT). Nevertheless, extending our evaluation to include DT is an interesting direction for future work!
>
> > Can the authors provide any intuition on the failure of Transformers and LSTM in the long-term credit assignment tasks?
>
> We find that in Active T-Maze task that has a credit assignment length of 250, the Transformer-based agent does not reach the Oracle (return lower than 1), but still reaches the Junction (return higher than 0). This indicates that Transformer-based agent may fail to explore enough to reach the Oracle, although the exploration should be relatively easy (just taking the initial action to move left). In LSTM-based agent, the return can *sometimes* be negative, indicating that it does not consistently reach the Junction and may face additional issues beyond credit assignment. Similar issues seem to occur in the Key-to-Door tasks where both agents do not reach the key in the initial phase.
>
> > It would be nice to see experimental evaluation of Transformers and LSTM in additional environments other than the toy tasks for better context.
>
> We agree with this limitation, and we plan to evaluate agents on more complicated environments in future work.

---

### Official Review · Reviewer_naYb · 2023-07-06

**Soundness:** 4 excellent
**Presentation:** 4 excellent
**Contribution:** 4 excellent
**Rating:** 8
**Confidence:** 2

**Summary:**

This work aims to answer how the parametrization of policy in RL affects the performance of sparse-reward tasks that require memory and credit assignment. In particular, the authors are interested in when transformers perform well. The authors define in the POMDP settings their own criteria of how to quantify policy memory, and credit assignment. The authors go through a list of previous benchmarks to clarify which benchmarks require which, and design their own experiment which decouples the two effects. The authors conclude that while transformes do well in pure memory tasks, they do not show better credit assignment capabilities compared to LSTMs.


**Strengths:**

- The question that was posed was quite unique and interesting.
- The authors do a convincing empirical study of which settings transformers perform better in, and the conclusions are aligned with empirical intuition.
- Conducted analysis of previous benchmarks is extensive, and the authors have a good experiment design.


**Weaknesses:**

- The motivations for memory length is reasonable as a quantiative metric, but I wonder if there are other quantitative criteria for judging memory and credit assignment. For policies with memory (i.e. recurrent), the definition seems less relevant since the policy can remember parts of all the information starting from time 0. On the other hand, the credit assignment length definition was novel and I found it quite relevant.


**Questions:**

- One of the key discussion that seem to be missing is whether or not the policy has the ability to succesfully abstract states that matter. For example, both in works along the lines of MDP compression with bisimulation, or Approximate Information States, they hypothesize a reduced-order model of the POMDP that the policy could possibly have that is sufficient for predicting rewards.
- I wonder if aside from credit assignment and history, if transformers are predicting states that truly matter for the task as well (which is partly a question of credit assignment, but less of a question of "when" something mattered.) For example, suppose a pixel-domain example that controls an agent, but the background constantly has something going on (e.g. humans may play tennis in the park when ducks are flying in the background). In this case excelling at pure memory is actually disadvantageous because the ducks are completely irrelevant to the task.
- In operations research there are classes of "submodular functions", and these have provably efficient greedy algorithms that are quantifiably suboptimal. I wonder how this is connected to credit assignment length.

**Limitations:**

The authors have adequately addressed limitations.

---

> ### Author Rebuttal · Authors · 2023-08-09
>
>
> > For policies with memory (i.e. recurrent), the definition seems less relevant since the policy can remember parts of all the information starting from time 0.
>
> Recurrent policies can *take* all the information starting from time 0 as inputs, but not necessarily *remember* it starting from that point.  As mentioned in L88-91, the context lengths of recurrent policies can be infinite, but their memory lengths can be very short, due to training issues such as gradient vanishing or explosion.  In this sense, our definition of memory lengths remains relevant to recurrent policies, with context lengths being infinite.
>
> > Lacking discussion on state and history abstraction, for example, bisimulation and approximate information states.
>
> Thank you for pointing it out. This work focuses on the architectural aspect (e.g., LSTMs and Transformers) of RL rather than the objective aspect (e.g., different abstraction methods). We view these two aspects as orthogonal and believe they can complement each other to enhance RL performance. We will include some related work on abstraction and clarify our focus in the camera-ready version.
>
> >  Are Transformers predicting states that truly matter for the task?
>
> We evaluate model-free RL agents equipped with memory architectures. Unlike model-based RL, which explicitly predicts next states, model-free RL purely aims to maximize returns. Therefore, we do not think Transformer-based RL agents used in our experiments face the issue of predicting irrelevant parts of states, regardless of their memory capability.
>
> > In operations research there are classes of "submodular functions", and these have provably efficient greedy algorithms that are quantifiably suboptimal. How is this connected to credit assignment length?
>
> After examining the definition of submodular functions, we don't see a direct connection between it and credit assignment length.  We are open to further insights or clarification on this connection, as it may lead to an intriguing avenue for future research.

---

> > ### Comment · Reviewer_naYb · 2023-08-14
> >
> > I'd like to thank the authors for the response, my scores are the same.

---

### Official Review · Reviewer_mXpQ · 2023-07-07

**Soundness:** 4 excellent
**Presentation:** 4 excellent
**Contribution:** 4 excellent
**Rating:** 7
**Confidence:** 3

**Summary:**

In this paper the authors consider the effectiveness of transformers for solving two kinds of RL problems: credit assignment and memory. To achieve their goal the authors develop rigorous definitions for memory and credit assignment so that different RL tasks can be understood and compared in these terms. The authors then review many well known RL problems from the literature and classify them according to their definitions. Going one step further the authors then propose a new RL task where the role of memory and credit assignment can be disambiguated. Finally, the authors perform empirical experiments on many of the previously analyzed RL tasks to understand when transformers shine and when they fall short.

**Strengths:**

The paper is accessible, rigorous, and timely. Great work.

**Weaknesses:**

1. If I were to name anything it would be I wish the authors could have provided a little more on what they think potential solutions could be to the credit assignment problem when using transformers. The paper stands on its own without this, but the authors have clearly thought about the problem and their insights could be valuable.

**Questions:**

1. I'm curious given the time and thought the authors have put into this problem what they think may be a solution. Can we simply slap on a specific submodule to transformers to augment their credit assignment capabilities? Or do you think we need to develop fundamentally new architectures?

**Limitations:**

The authors have adequately addressed limitations.

---

> ### Author Rebuttal · Authors · 2023-08-09
>
> > What potential solutions could be to the credit assignment problem when using Transformers?
>
> We think the credit assignment problem might be easier to solve with Transformers by using them to explicitly redistribute reward signals [Liu et al., 2019, Ferret et al., 2020]. Specifically, Transformers can efficiently redistribute distant reward signals to the time step when the corresponding action occurs. Their strength in handling long-term memory makes them suitable for learning such temporal dependencies with appropriate objectives beyond model-free RL.
>
> > Can we simply slap on a specific submodule to Transformers to augment their credit assignment capabilities? Or do you think we need to develop fundamentally new architectures?
>
> We conduct new experiments on *multi-layer* Transformers in credit assignment tasks. Although they aid in medium-term credit assignment, they do not help with long-term credit assignment. Please see the **general response** for details. This result suggests that simply adding a specific submodule to Transformers is unlikely to improve long-term credit assignment, indicating the potential need for new architectures.
>
> Please let us know if there are any further questions.

---

> > ### Comment · Reviewer_mXpQ · 2023-08-14
> >
> > Thank you for your responses. I remain positive about the paper.

---

### Author Rebuttal · Authors · 2023-08-09

# General Response

First, we would like to thank all four reviewers for their positive and constructive feedback on our work!
During the rebuttal period, we conducted additional experiments to address some of the questions posed by the reviewers.

**mXpQ  (see Figure 1 in the PDF)**

> Can we simply slap on a specific submodule to Transformers to augment their credit assignment capabilities?

We showed that (single-layer) Transformers cannot help long-term credit assignment. To further confirm this conclusion, we ran *multi-layer* Transformers on these tasks.
In both Active T-Maze and Key-to-Door, we find that both 2-layer and 4-layer Transformers greatly outperform 1-layer Transformer in tasks with medium-term credit assignment (length $\le 200$), while still fail to perform long-term credit assignment (length $\ge 250$).
This finding suggests that it is *unlikely* that simply increasing the size of Transformers will enable them to excel in long-term credit assignment.

**2AdR (see Figure 2 in the PDF)**

> LSTM starts to falter at a memory length of 250 for Passive T-Maze. But Fig 2 shows that LSTM can solve it with a memory length of 750. This suggests that hyperparameter tuning might influence LSTM training. Will the results of LSTM be improved by further tuning?

First, we want to clarify that although LSTM reaches returns higher than $0.5$ in Passive T-Maze with a memory length of $750$, it does not *solve* the task, requiring the return to be $1.0$.

Your point on tuning may be valid for medium-term memory lengths. But for long-term memory tasks, our new results show that tuning is unlikely to help LSTM.
During the rebuttal period, we conduct new experiments on LSTM-based agent on the tasks with memory lengths of 1250 and 1500. LSTM-based agent reaches return below 0.5.
The results confirm our conclusion that LSTM-based agent cannot solve long-term memory tasks.

**2AdR**

> Can you do a significance test to show superiority of GPT2 over LSTM in Figure 4?

Following the reviewer's suggestion, we did a Welch’s t-test on Passive Visual Match success rates (Figure 4 in the main paper). The p-values for the memory length of 500 and 750 are 0.698 and 0.038, respectively.
These results indicate that there is significant evidence to reject the null hypothesis at the 750 memory length, but not at 500. Therefore, we clarify that the advantages observed with GPT at a memory length of 500 predominantly pertain to short-term memory, rather than long-term memory.

---

### Decision · Program_Chairs · 2023-09-21

**Decision:**

Accept (oral)

**Comment:**

This paper aims to understand the success of transformer-based methods for reinforcement learning. In particular, they aim to understand the role of transformers in addressing two challenges (1) credit assignment (understanding how actions influence future rewards), and (2) memory (representing prior observations); both challenges entail modeling long-term dependencies. To separate these challenges, the authors design a suite of experimental benchmarks, and find transformers are effective at enhancing memory, but do not improve credit assignment.

Reviewers found the paper's topic to be timely, and found the results to be convincing and likely to be of significant interest to the reinforcement learning community. In addition, they found the paper to be accessible and well-written, and unanimously advocated for its acceptance.